# RETHINKING MODEL ENSEMBLE IN TRANSFER-BASED ADVERSARIAL ATTACKS

**Huanran Chen[1,2], Yichi Zhang[2,3], Yinpeng Dong[2,3*], Xiao Yang[2], Hang Su[2], Jun Zhu[2,3*]**
[1]School of Computer Science, Beijing Institute of Technology
[2]Dept. of Comp. Sci. and Tech., Institute for AI, Tsinghua-Bosch Joint ML Center, THBI Lab,
BNRist Center, Tsinghua University, Beijing, 100084, China    [3]RealAI
huanranchen@bit.edu.cn, {zyc22,yangxiao19}@mails.tsinghua.edu.cn
{dongyinpeng, suhangss, dcszj}@mail.tsinghua.edu.cn

## ABSTRACT

It is widely recognized that deep learning models lack robustness to adversarial examples. An intriguing property of adversarial examples is that they can transfer across different models, which enables black-box attacks without any knowledge of the victim model. An effective strategy to improve the transferability is attacking an ensemble of models. However, previous works simply average the outputs of different models, lacking an in-depth analysis on how and why model ensemble methods can strongly improve the transferability. In this paper, we rethink the ensemble in adversarial attacks and define the common weakness of model ensemble with two properties: 1) the flatness of loss landscape; and 2) the closeness to the local optimum of each model. We empirically and theoretically show that both properties are strongly correlated with the transferability and propose a Common Weakness Attack (CWA) to generate more transferable adversarial examples by promoting these two properties. Experimental results on both image classification and object detection tasks validate the effectiveness of our approach to improving the adversarial transferability, especially when attacking adversarially trained models. We also successfully apply our method to attack a black-box large vision-language model – Google's Bard, showing the practical effectiveness. Code is available at https://github.com/huanranchen/AdversarialAttacks.

## 1 INTRODUCTION

Deep learning has achieved remarkable progress over the past decade and has been widely deployed in real-world applications (LeCun et al., 2015). However, deep neural networks (DNNs) are vulnerable to adversarial examples (Szegedy et al., 2014; Goodfellow et al., 2015), which are crafted by imposing human-imperceptible perturbations to natural examples. Adversarial examples can mislead the predictions of the victim model, posing great threats to the security of DNNs and their applications (Kurakin et al., 2018; Sharif et al., 2016). The study of generating adversarial examples, known as adversarial attack, has attracted tremendous attention since it can provide a better understanding of the working mechanism of DNNs (Dong et al., 2017), evaluate the robustness of different models (Carlini & Wagner, 2017), and help to design more robust and reliable algorithms (Madry et al., 2018).

Adversarial attacks can be generally categorized into white-box and black-box attacks according to the adversary's knowledge of the victim model. With limited model information, black-box attacks either rely on query feedback (Chen et al., 2017) or leverage the transferability (Liu et al., 2017) to generate adversarial examples. Particularly, transfer-based attacks use adversarial examples generated for white-box surrogate models to mislead black-box models, which do not demand query access to the black-box models, making them more practical in numerous real-world scenarios (Liu et al., 2017; Dong et al., 2018). With the development of adversarial defenses (Madry et al., 2018; Tramèr et al., 2018; Wong et al., 2020; Wei et al., 2023b) and diverse network architectures (Dosovitskiy et al., 2020; Liu et al., 2021; 2022b), the transferability of existing methods can be largely degraded.

---

*Y. Dong and J. Zhu are corresponding authors. H. Chen has done this work at Tsinghua University.

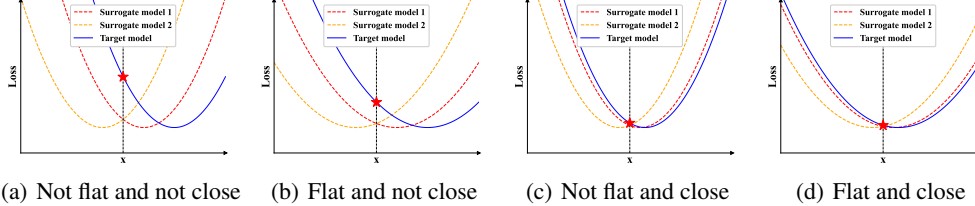

Figure 1: **Illustration of Common Weakness.** The generalization error is strongly correlated with the flatness of loss landscape and the distance between the solution and the closest local optimum of each model. We define the common weakness of model ensemble as the solution that is at the flat landscape and close to local optima of training models, as shown in (d).

By making an analogy between the transferability of adversarial examples and the generalization of deep neural networks (Dong et al., 2018), many researchers improve the transferability by designing advanced optimization algorithms to avoid undesirable optimum (Dong et al., 2018; Lin et al., 2020; Wang & He, 2021; Wang et al., 2021) or leveraging data augmentation strategies to prevent overfitting (Xie et al., 2019; Dong et al., 2019; Lin et al., 2020). Meanwhile, generating adversarial examples for multiple surrogate models can further improve the transferability, similar to training on more data to improve model generalization (Dong et al., 2018). Early approaches simply average the outputs of multiple models in loss (Liu et al., 2017) or in logits (Dong et al., 2018), but ignore their different properties. Xiong et al. (2022) introduce the SVRG optimizer to reduce the variance of gradients of different models during optimization. Nevertheless, recent studies (Yang et al., 2021) demonstrate that the variance of gradients does not always correlate with the generalization performance. From the model perspective, increasing the number of surrogate models can reduce the generalization error (Huang et al., 2023), and some studies (Li et al., 2020; Huang et al., 2023) propose to create surrogate models from existing ones. However, the number of surrogate models in adversarial attacks is usually limited due to the high computational costs, making it necessary to develop new algorithms that can leverage fewer surrogate models while achieving improved attack success rates.

In this paper, we rethink the ensemble methods in adversarial attacks. By using a quadratic approximation of the expected attack objective over the victim models, we observe that the second-order term involves the Hessian matrix of the loss function and the squared $\ell_2$ distance to the local optimum of each model, both of which are strongly correlated with the adversarial transferability (as illustrated in Fig. 1), especially when there are only a few models in the ensemble. Based on these two terms, we define the **common weakness** of an ensemble of models as the solution that is at the flat landscape and close to the models' local optima. To generate adversarial examples that exploit the common weakness of model ensemble, we propose a **Common Weakness Attack (CWA)** composed of two sub-methods named Sharpness Aware Minimization (SAM) and Cosine Similarity Encourager (CSE), which are designed to optimize the two properties of common weakness separately. As our methods are orthogonal to previous methods, *e.g.*, MI (Dong et al., 2018), VMI (Wang & He, 2021), SSA (Long et al., 2022), our methods can be incorporated with them seamlessly for improved performance.

We conduct extensive experiments to confirm the superior transferability of adversarial examples generated by our methods. We first verify in image classification for 31 victim models with various architectures (*e.g.*, CNNs (He et al., 2016), Transformers (Dosovitskiy et al., 2020; Liu et al., 2021)) and training settings (*e.g.*, standard training, adversarial training (Salman et al., 2020; Wong et al., 2020), input purification (Naseer et al., 2020; Nie et al., 2022)). Impressively, when attacking the state-of-the-art defense models, MI-CWA surpasses MI with the logits ensemble strategy (Dong et al., 2018) by 30% under the black-box setting. We extend our method to object detection by generating a universal adversarial patch that achieves an averaged mAP of 9.85 over 8 modern detectors, surpassing the recent work (Huang et al., 2023). Ablation studies also validate that our methods indeed flatten the loss landscape and increase the cosine similarity of gradients of different models. Moreover, we demonstrate successful black-box attacks against a recent large vision-language model (*i.e.*, Google's Bard) based on our method, outperforming the baseline method by a large margin.

## 2 RELATED WORK

In this section, we briefly review the existing transfer-based adversarial attack algorithms. Overall, these methods usually make an analogy between the transferability of adversarial examples and the generalization of deep learning models, and we categorize them into 3 classes as follows.

**Gradient-based optimization.** Comparing the optimization of adversarial examples and the training of deep learning models, researchers introduce some optimization techniques to boost the transferability of adversarial examples. Momentum Iterative (MI) method (Dong et al., 2018) and Nesterov Iterative (NI) method (Lin et al., 2020) introduce the momentum optimizer and Nesterov accelerated gradient to prevent the adversarial examples from falling into undesired local optima. Enhanced Momentum Iterative (EMI) method (Wang et al., 2021) accumulates the average gradient of the data points sampled in the gradient direction of the previous iteration to stabilize the update direction and escape from poor local maxima. Variance-tuning Momentum Iterative (VMI) method (Wang & He, 2021) reduces the variance of the gradient during the optimization via tuning the current gradient with the gradient variance in the neighborhood of the previous data point.

**Input transformation.** These methods transform the input images before feeding to the classifier for diversity, similar to data augmentation techniques in deep learning. Diverse Inputs (DI) method (Xie et al., 2019) applies random resizing and padding to the input images. Translation-Invariant (TI) attack (Dong et al., 2019) derives an efficient algorithm for calculating the gradients w.r.t. the translated images, which is equivalent to applying translations to the input images. Scale-Invariant (SI) attack (Lin et al., 2020) scales the images with different factors based on the observation that the model has similar performance on these scaled images.

**Model ensemble methods.** As mentioned in Huang et al. (2023), increasing the number of classifiers in adversarial attacks can reduce the generalization error upper bound in Empirical Risk Minimization (ERM), similar to increasing the number of training examples in deep learning. Dong et al. (2018) propose to alternatively average over losses, predicted probabilities, or logits of surrogate models to form an ensemble. Li et al. (2020) and Huang et al. (2023) also propose novel methods to generate massive variants of the surrogate models and then take the average value of losses or logits. Xiong et al. (2022) introduce the SVRG optimizer (Allen-Zhu & Yuan, 2016) into ensemble adversarial attack to reduce the variance of the gradients during optimization. However, the number of surrogate models in practice is usually small, which cannot guarantee a satisfying generalization upper bound in the theory of ERM, and only focusing on variance does not necessarily lead to better generalization (Yang et al., 2021). Additionally, self-ensembled surrogate models typically exhibit lower effectiveness compared to standard surrogate models, resulting in limited performance.

## 3 METHODOLOGY

In this section, we present our formulation of common weakness based on a second-order approximation and propose the **Common Weakness Attack (CWA)** composed of Sharp Aware Minimization (SAM) and Cosine Similarity Encourager (CSE).

### 3.1 PRELIMINARIES

We let $\mathcal{F} := \{f\}$ denote the set of possible image classifiers for a given task, where each classifier $f : \mathbb{R}^D \to \mathbb{R}^K$ outputs the logits over $K$ classes for an input $\boldsymbol{x} \in \mathbb{R}^D$. Given a natural image $\boldsymbol{x}_{nat}$ and the corresponding label $y$, transfer-based attacks aim to craft an adversarial example $\boldsymbol{x}$ that could be misclassified by the models in $\mathcal{F}$. It can be formulated as a constrained optimization problem:

$$\min_{\boldsymbol{x}} \mathbb{E}_{f \in \mathcal{F}}[L(f(\boldsymbol{x}), y)], \text{ s.t. } \|\boldsymbol{x} - \boldsymbol{x}_{nat}\|_\infty \leq \epsilon, \tag{1}$$

where $L$ is the loss function, *e.g.*, negative cross-entropy loss as $L(f(\boldsymbol{x}), y) = \log(\text{softmax}(f(\boldsymbol{x}))_y)$, and we study the $\ell_\infty$ norm. Eq. (1) can be approximated using a few "training" classifiers $\mathcal{F}_t := \{f_i\}_{i=1}^n \subset \mathcal{F}$ (known as ensemble) as $\frac{1}{n}\sum_{i=1}^n L(f_i(\boldsymbol{x}), y)$ (*i.e.*, loss ensemble (Liu et al., 2017)) or $L(\frac{1}{n}\sum_{i=1}^n f_i(\boldsymbol{x}), y)$ (*i.e.*, logits ensemble (Dong et al., 2018)). Previous works (Dong et al., 2018; 2019; Xie et al., 2019; Wang & He, 2021) propose gradient computation or input transformation methods based on this empirical loss for better transferability. For example, the momentum iterative (MI) method (Dong et al., 2018) performs gradient update (as shown in Fig. 2(a)) as

$$\boldsymbol{m} = \mu \cdot \boldsymbol{m} + \frac{\nabla_{\boldsymbol{x}} L(\frac{1}{n}\sum_{i=1}^n f_i(\boldsymbol{x}), y)}{\|\nabla_{\boldsymbol{x}} L(\frac{1}{n}\sum_{i=1}^n f_i(\boldsymbol{x}), y)\|_1}; \quad \boldsymbol{x}_{t+1} = \text{clip}_{\boldsymbol{x}_{nat}, \epsilon}(\boldsymbol{x}_t + \alpha \cdot \text{sign}(\boldsymbol{m})),$$

where $\boldsymbol{m}$ accumulates the gradients, $\mu$ is the decay factor, $\alpha$ is the step size, $\text{clip}_{\boldsymbol{x}_{nat}, \epsilon}(\boldsymbol{x})$ would project $\boldsymbol{x}$ to the $\ell_\infty$ ball around $\boldsymbol{x}_{nat}$ with radius $\epsilon$, and we use the logits ensemble strategy here.

## 3.2 MOTIVATION OF COMMON WEAKNESS

Although the existing methods can improve the transferability to some extent, a recent work (Huang et al., 2023) points out that the optimization of adversarial example conforms to Empirical Risk Minimization (ERM) (Vapnik, 1999) and the limited number of training models could lead to a large generalization error. In this paper, we consider a quadratic approximation of the objective in Eq. (1).

Formally, we let $\boldsymbol{p}_i$ denote the closest optimum of model $f_i \in \mathcal{F}$ to $\boldsymbol{x}$ and $\boldsymbol{H}_i$ denote the Hessian matrix of $L(f_i(\boldsymbol{x}), y)$ at $\boldsymbol{p}_i$. We employ the second-order Taylor expansion to approximate Eq. (1) at $\boldsymbol{p}_i$ for each model $f_i$ as

$$\mathbb{E}_{f_i \in \mathcal{F}} \left[ L(f_i(\boldsymbol{p}_i), y) + \frac{1}{2}(\boldsymbol{x} - \boldsymbol{p}_i)^\top \boldsymbol{H}_i(\boldsymbol{x} - \boldsymbol{p}_i) \right]. \tag{2}$$

In the following, we omit the subscript of the expectation in Eq. (2) for simplicity. Based on Eq. (2), we can see that a smaller value of $\mathbb{E}[L(f_i(\boldsymbol{p}_i), y)]$ and $\mathbb{E}[(\boldsymbol{x} - \boldsymbol{p}_i)^\top \boldsymbol{H}_i(\boldsymbol{x} - \boldsymbol{p}_i)]$ means a smaller loss on testing models, *i.e.*, better transferability. The first term represents the loss value of each model at its own optimum $\boldsymbol{p}_i$. Some previous works (Laurent & Brecht, 2018; Kawaguchi et al., 2019) have proven that the local optima have nearly the same value with the global optimum in neural networks. Consequently, there may be no room for further improving this term, and we put our main focus on the second term to achieve better transferability.

In the following theorem, we derive an upper bound for the second term, since directly optimizing it which requires third-order derivative is intractable.

**Theorem 3.1.** *(Proof in Appendix A.1) Assume that the covariance between $\|\boldsymbol{H}_i\|_F$ and $\|\boldsymbol{p}_i - \boldsymbol{x}\|_2$ is zero, we can get the upper bound of the second term as*

$$\mathbb{E}[(\boldsymbol{x} - \boldsymbol{p}_i)^\top \boldsymbol{H}_i(\boldsymbol{x} - \boldsymbol{p}_i)] \leq \mathbb{E}[\|\boldsymbol{H}_i\|_F]\mathbb{E}[\|(\boldsymbol{x} - \boldsymbol{p}_i)\|_2^2]. \tag{3}$$

Intuitively, $\|\boldsymbol{H}_i\|_F$ represents the sharpness/flatness of loss landscape (Tu et al., 2016; Tsuzuku et al., 2020; Liu et al., 2022a; Wei et al., 2023c) and $\|(\boldsymbol{x} - \boldsymbol{p}_i)\|_2^2$ describes the translation of landscape. The two terms can be assumed independent and therefore their covariance can be assumed as zero. As shown in Theorem 3.1, a smaller value of $\mathbb{E}[\|\boldsymbol{H}_i\|_F]$ and $\mathbb{E}[\|(\boldsymbol{x} - \boldsymbol{p}_i)\|_2^2]$ provides a smaller bound and further leads to a smaller loss on testing models. Previous works have pointed out that a smaller norm of the Hessian matrix indicates a flatter landscape of the objective, which is strongly correlated with better generalization (Wu et al., 2017; Li et al., 2018; Chen et al., 2022). As for $\mathbb{E}[\|(\boldsymbol{x} - \boldsymbol{p}_i)\|_2^2]$ representing the squared $\ell_2$ distance from $\boldsymbol{x}$ to the closest optimum $\boldsymbol{p}_i$ of each model, we empirically show that it also has a tight connection with generalization and adversarial transferability in Appendix D.2 and theoretically prove this under a simple case in Appendix A.2. Fig. 1 provides an illustration that the flatness of loss landscape and closeness between local optima of different models can help with adversarial transferability.

Based on the analysis above, the optimization of an adversarial example turns into looking for a point near the optima of victim models (*i.e.*, minimizing the original objective), while pursuing that the landscape at the point is flat and the distance from it to each optimum is close. The latter two targets are our main findings and we here define the concept of **common weakness** with these two terms as a local point $\boldsymbol{x}$ who has a small value of $\mathbb{E}[\|\boldsymbol{H}_i\|_F]$ and $\mathbb{E}[\|(\boldsymbol{x} - \boldsymbol{p}_i)\|_2^2]$. Note that there is no clear boundary between common weakness and non-common weakness – the smaller these two terms are, the more likely $\boldsymbol{x}$ to be the common weakness. The ultimate goal is to find an example that has the properties of common weakness. As Theorem 3.1 indicates, we can achieve this by optimizing these two terms separately.

## 3.3 SHARPNESS AWARE MINIMIZATION

To cope with the flatness of loss landscape, we minimize $\|\boldsymbol{H}_i\|_F$ for each model in the ensemble. However, this requires third-order derivative w.r.t. $\boldsymbol{x}$, which is computationally expensive. There are some research to ease the sharpness of loss landscape in model training (Foret et al., 2020; Izmailov et al., 2018; Kwon et al., 2021). The Sharpness Aware Minimization (SAM) (Foret et al., 2020) is an effective algorithm to acquire a flatter landscape, which is formulated as a min-max optimization problem. The inner maximization aims to find a direction along which the loss changes more rapidly; while the outer problem minimizes the loss at this direction to improve the flatness of loss landscape.

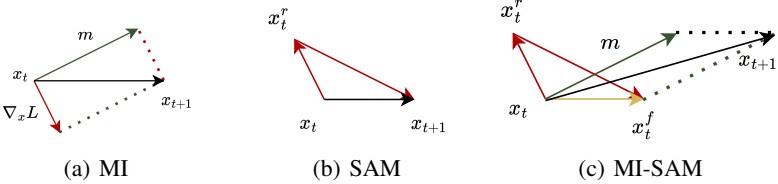

|     |     |     |
| --- | --- | --- |
| (a) MI | (b) SAM | (c) MI-SAM |

Figure 2: Illustration of MI, SAM, and MI-SAM. The symbols are introduced in Eq. (4)-(6)
.

In the context of generating adversarial examples restricted by the $\ell_\infty$ norm as in Eq. (1), we aim to optimize the flatness of landscape in the space of $\ell_\infty$ norm to boost the transferability, which is different from the original SAM in the space of $\ell_2$ norm. Therefore, we derive a modified SAM algorithm suitable for the $\ell_\infty$ norm (in Appendix B.1). As shown in Fig. 2, at the $t$-th iteration of adversarial attacks, the SAM algorithm first performs an inner gradient ascent step at the current adversarial example $\boldsymbol{x}_t$ with a step size $r$ as

$$\boldsymbol{x}_t^r = \text{clip}_{\boldsymbol{x}_{nat},\epsilon}\left(\boldsymbol{x}_t + r \cdot \text{sign}\left(\nabla_{\boldsymbol{x}} L\left(\frac{1}{n}\sum_{i=1}^n f_i(\boldsymbol{x}_t), y\right)\right)\right), \tag{4}$$

and then performs an outer gradient descent step at $\boldsymbol{x}_t^r$ with a step size $\alpha$ as

$$\boldsymbol{x}_t^f = \text{clip}_{\boldsymbol{x}_{nat},\epsilon}\left(\boldsymbol{x}_t^r - \alpha \cdot \text{sign}\left(\nabla_{\boldsymbol{x}} L\left(\frac{1}{n}\sum_{i=1}^n f_i(\boldsymbol{x}_t^r), y\right)\right)\right). \tag{5}$$

Note that in Eq. (4) and Eq. (5), we apply SAM on the model ensemble rather than each training model. Thus, we can not only perform parallel computation during the backward pass to improve efficiency, but also obtain better results using the logits ensemble strategy (Dong et al., 2018).

Moreover, we can combine the reverse step and forward step of SAM as a single update direction $\boldsymbol{x}_t^f - \boldsymbol{x}_t$, and integrate it into existing attack algorithms. For example, the integration of MI and SAM yields the MI-SAM algorithm with the following update (as shown in Fig. 2(c))

$$\boldsymbol{m} = \mu \cdot \boldsymbol{m} + \boldsymbol{x}_t^f - \boldsymbol{x}_t; \quad \boldsymbol{x}_{t+1} = \text{clip}_{\boldsymbol{x}_{nat},\epsilon}(\boldsymbol{x}_t + \boldsymbol{m}), \tag{6}$$

where $\boldsymbol{m}$ accumulates the gradients with a decay factor $\mu$. By iteratively repeating this procedure, the adversarial example will converge into a flatter loss landscape, leading to better transferability.

### 3.4 COSINE SIMILARITY ENCOURAGER

We then develop an algorithm to make the adversarial example converge to a point which is close to the local optimum of each model. Instead of directly optimizing $\frac{1}{n}\sum_{i=1}^n \|(\boldsymbol{x} - \boldsymbol{p}_i)\|_2^2$ with the training models in the ensemble, since it is hard to compute the gradient w.r.t. $\boldsymbol{x}$, we derive an upper bound for this loss.

**Theorem 3.2.** *(Proof in Appendix A.3) The upper bound of $\frac{1}{n}\sum_{i=1}^n \|(\boldsymbol{x} - \boldsymbol{p}_i)\|_2^2$ is proportional to the dot product similarity between the gradients of all models:*

$$\frac{1}{n}\sum_{i=1}^n \|(\boldsymbol{x} - \boldsymbol{p}_i)\|_2^2 \leq -\frac{2M}{n}\sum_{i=1}^n\sum_{j=1}^{i-1} \boldsymbol{g}_i \boldsymbol{g}_j, \tag{7}$$

*where $M = \max \|\boldsymbol{H}_i^{-1}\|_F^2$ and $\boldsymbol{g}_i = \nabla_{\boldsymbol{x}} L(f_i(\boldsymbol{x}), y)$ represents the gradient of the $i$-th model.*

Based on Theorem 3.2, the minimization of the distance to the local optimum of each model turns into the maximization of the dot product between gradients of different models. To solve this problem, Nichol et al. (2018) propose an efficient algorithm via first-order derivative approximation. We apply this algorithm to generate adversarial examples by successively performing gradient updates using each model $f_i$ sampled from the ensemble $\mathcal{F}_t$ with a small step size $\beta$. The update process is

$$\boldsymbol{x}_t^i = \text{clip}_{\boldsymbol{x}_{nat},\epsilon}(\boldsymbol{x}_t^{i-1} - \beta \cdot \nabla_{\boldsymbol{x}} L(f_i(\boldsymbol{x}_t^{i-1}), y)), \tag{8}$$

where $\boldsymbol{x}_t^0 = \boldsymbol{x}_t$. Once the update for every model is complete, we calculate the final update using a larger step size $\alpha$ as

$$\boldsymbol{x}_{t+1} = \text{clip}_{\boldsymbol{x}_{nat},\epsilon}(\boldsymbol{x}_t + \alpha \cdot (\boldsymbol{x}_t^n - \boldsymbol{x}_t)). \tag{9}$$

Although directly applying this algorithm can achieve good results, it is incompatible with SAM due to the varying scales of gradient norm (Dong et al., 2018). To solve this problem, we normalize the

---

**Algorithm 1** MI-CWA algorithm

---

**Require:** natural image $\boldsymbol{x}_{nat}$, label $y$, perturbation budget $\epsilon$, iterations $T$, loss function $L$, model ensemble $\mathcal{F}_t = \{f_i\}_{i=1}^n$, decay factor $\mu$, step sizes $r$, $\beta$ and $\alpha$.
1: **Initialize:** $\boldsymbol{m} = 0$, inner momentum $\hat{\boldsymbol{m}} = 0$, $\boldsymbol{x}_0 = \boldsymbol{x}_{nat}$;
2: **for** $t = 0$ **to** $T - 1$ **do**
3:     Calculate $\boldsymbol{g} = \nabla_{\boldsymbol{x}} L(\frac{1}{n} \sum_{i=1}^n f_i(\boldsymbol{x}_t), y)$;
4:     Update $\boldsymbol{x}_t$ by $\boldsymbol{x}_t^0 = \text{clip}_{\boldsymbol{x}_{nat}, \epsilon}(\boldsymbol{x}_t + r \cdot \text{sign}(\boldsymbol{g}))$;
5:     **for** $i = 1$ **to** $n$ **do**
6:         Calculate $\boldsymbol{g} = \nabla_{\boldsymbol{x}} L(f_i(\boldsymbol{x}_t^{i-1}), y)$;
7:         Update inner momentum by $\hat{\boldsymbol{m}} = \mu \cdot \hat{\boldsymbol{m}} + \frac{\boldsymbol{g}}{\|\boldsymbol{g}\|_2}$;
8:         Update $\boldsymbol{x}_t^i$ by $\boldsymbol{x}_t^i = \text{clip}_{\boldsymbol{x}_{nat}, \epsilon}(\boldsymbol{x}_t^{i-1} - \beta \cdot \hat{\boldsymbol{m}})$;
9:     **end for**
10:    Calculate the update $\boldsymbol{g} = \boldsymbol{x}_t^n - \boldsymbol{x}_t$;
11:    Update momentum $\boldsymbol{m} = \mu \cdot \boldsymbol{m} + \boldsymbol{g}$;
12:    update $\boldsymbol{x}_{t+1}$ by $\boldsymbol{x}_{t+1} = \text{clip}_{\boldsymbol{x}_{nat}, \epsilon}(\boldsymbol{x}_t + \alpha \cdot \text{sign}(\boldsymbol{m}))$;
13: **end for**
14: **Return:** $\boldsymbol{x}_T$.

---

gradient at each update by their $\ell_2$ norm. We discover that the modified version actually maximizes the cosine similarity between gradients (proof in Appendix B.2). Thus, we call it Cosine Similarity Encourager (CSE), which can be further combined with MI as MI-CSE. MI-CSE involves an inner momentum to accumulate the gradients of each model. We provide the pseudocode in Appendix B.2.

### 3.5 COMMON WEAKNESS ATTACK

Given the respective algorithms to optimize the flatness of loss landscape and the closeness between local optima of different models, we thus need to combine them as a unified Common Weakness Attack (CWA) to achieve better transferability. In consideration of the feasibility of parallel gradient backpropagation and time complexity, we substitute the second step of SAM with CSE, and the resulting algorithm is called CWA. We also combine CWA with MI to obtain MI-CWA, with the psuedocode shown in Algorithm 1. CWA can also be incorporated with other strong adversarial attack algorithms, including VMI (Wang & He, 2021), SSA (Long et al., 2022), *etc*. The details of these algorithms are provided in Appendix B.3.

## 4 EXPERIMENTS

In this section, we conduct comprehensive experiments to show the superiority of our methods. The tasks range from image classification to object detection and image description with the recent large vision-language models, demonstrating the universality of our methods.

### 4.1 ATTACK IN IMAGE CLASSIFICATION

**Experimental settings. (1) Dataset:** Similar to previous works, we adopt the NIPS2017 dataset[1], which is comprised of 1000 images compatible with ImageNet (Russakovsky et al., 2015). All the images are resized to $224 \times 224$. **(2) Surrogate models:** We choose four normally trained models – ResNet-18, ResNet-32, ResNet-50, ResNet-101 (He et al., 2016) from TorchVision (Marcel & Rodriguez, 2010), and two adversarially trained models – ResNet-50 (Salman et al., 2020), XCiT-S12 (Debenedetti et al., 2023) from RobustBench (Croce et al., 2021). They contain both normal and robust models, which is effective to assess the attacker's ability to utilize diverse surrogate models. We further conduct experiments with other surrogate models and other settings (*e.g.*, using less diverse surrogates, settings in Dong et al. (2018), settings in Naseer et al. (2020), attacks under $\epsilon = 4/255$) in Appendix C. **(3) Black-box models:** We choose a model from each model type available in TorchVision and RobustBench as the black-box model. We also include 7 other prominent defenses in our evaluation. In total, there are 31 models selected. For further details, please refer to Appendix C.1. **(4) Compared methods:** We integrate our methods into MI (Dong et al., 2018) as MI-SAM, MI-CSE and MI-CWA. We also integrate CWA into two state-of-the-art methods VMI (Wang & He, 2021) and SSA (Long et al., 2022) as VMI-CWA and SSA-CWA. We compare them with FGSM (Goodfellow

---

[1]https://www.kaggle.com/competitions/nips-2017-non-targeted-adversarial-attack

Table 1: **Black-box attack success rate (%, ↑) on NIPS2017 dataset.** While our methods lead the performance on 16 normally trained models with various architectures, they significantly empower the transfer-based attack with great margins on 8 adversarially trained models available on RobustBench (Croce et al., 2021). The model details are provided in Appendix C.1.

| Method | Backbone | FGSM | BIM | MI | DI | TI | VMI | SVRE | PI | SSA | RAP | MI-SAM | MI-CSE | MI-CWA | VMI-CWA | SSA-CWA |
|---|---|---|---|---|---|---|---|---|---|---|---|---|---|---|---|---|
| Normal | AlexNet | 76.4 | 54.9 | 73.2 | 78.9 | 78.0 | 83.3 | 82.5 | 78.2 | 89.0 | 82.9 | 81.0 | 93.6 | 94.6 | 95.9 | **96.9** |
| | VGG-16 | 68.9 | 86.1 | 91.9 | 92.9 | 82.5 | 94.8 | 96.4 | 93.1 | 97.7 | 93.1 | 95.6 | 99.6 | 99.5 | **99.9** | 99.9 |
| | GoogleNet | 54.4 | 76.6 | 89.1 | 92.0 | 77.8 | 94.2 | 95.7 | 91.0 | 97.2 | 90.4 | 94.4 | 98.8 | 99.0 | **99.8** | 99.8 |
| | Inception-V3 | 54.5 | 64.9 | 84.6 | 89.0 | 75.7 | 91.1 | 92.6 | 85.9 | 95.6 | 85.0 | 89.2 | 97.3 | 97.2 | 98.9 | **99.6** |
| | ResNet-152 | 54.5 | 96.0 | 96.6 | 93.8 | 87.8 | 97.1 | 99.0 | 97.2 | 97.6 | 95.3 | 97.9 | 99.9 | 99.8 | **100.0** | 100.0 |
| | DenseNet-121 | 57.4 | 93.0 | 95.8 | 93.8 | 88.0 | 96.6 | 99.1 | 96.9 | 98.2 | 94.1 | 98.0 | 99.9 | 99.8 | 99.9 | **100.0** |
| | SqueezeNet | 85.0 | 80.4 | 89.4 | 92.9 | 85.8 | 94.2 | 96.1 | 92.1 | 97.2 | 92.1 | 94.1 | 99.1 | 99.3 | 99.6 | **99.8** |
| | ShuffleNet-V2 | 81.2 | 65.3 | 79.9 | 85.7 | 78.2 | 89.9 | 90.3 | 85.8 | 93.9 | 89.3 | 87.9 | 97.2 | 97.3 | 98.7 | **98.8** |
| | MobileNet-V3 | 58.9 | 55.6 | 71.8 | 78.6 | 74.5 | 87.3 | 80.6 | 77.1 | 91.4 | 81.1 | 80.7 | 94.6 | 95.7 | 97.8 | **98.1** |
| | EfficientNet-B0 | 50.8 | 80.2 | 90.1 | 91.5 | 76.8 | 94.6 | 96.7 | 93.3 | 96.9 | 91.4 | 95.2 | 98.8 | 98.9 | 99.7 | **99.9** |
| | MNasNet | 64.1 | 80.8 | 88.8 | 91.5 | 75.5 | 94.1 | 94.2 | 90.3 | 97.2 | 92.5 | 94.3 | 99.1 | 98.7 | 99.6 | **99.9** |
| | RegNetX-400MF | 57.1 | 81.1 | 89.3 | 91.2 | 82.4 | 95.3 | 95.4 | 91.0 | 97.4 | 90.8 | 93.9 | 98.9 | 99.4 | 99.8 | **99.9** |
| | ConvNeXt-T | 39.8 | 68.6 | 81.6 | 85.4 | 56.2 | 92.4 | 88.2 | 85.7 | 93.1 | 86.8 | 90.1 | 96.2 | 95.4 | 97.8 | **98.1** |
| | ViT-B/16 | 33.8 | 35.0 | 59.2 | 66.8 | 56.9 | 81.8 | 65.8 | 64.5 | 83.0 | 66.7 | 68.9 | 89.6 | 89.6 | **92.3** | 90.0 |
| | Swin-S | 34.0 | 48.2 | 66.0 | 74.2 | 40.9 | 84.2 | 73.4 | 69.1 | 85.2 | 72.2 | 75.1 | 88.6 | 87.6 | **91.6** | 88.4 |
| | MaxViT-T | 31.3 | 49.7 | 66.1 | 73.2 | 32.7 | 83.5 | 71.1 | 70.1 | 85.2 | 69.7 | 75.6 | 85.8 | 85.9 | **88.1** | 86.1 |
| FGSMAT | Inception-V3 | 53.9 | 43.4 | 55.9 | 61.8 | 66.1 | 72.3 | 66.8 | 61.1 | 84.3 | 69.6 | 64.5 | 89.6 | 89.6 | 91.5 | **92.7** |
| EnsAT | IncRes-V2 | 32.5 | 28.5 | 42.5 | 52.9 | 58.5 | 66.4 | 46.8 | 45.3 | 76.1 | 48.6 | 47.9 | 78.2 | 79.1 | 83.2 | **84.1** |
| FastAT | ResNet-50 | 45.6 | 41.6 | 45.7 | 47.1 | 49.3 | 51.4 | 51.0 | 33.1 | 34.7 | 56.5 | 50.6 | **75.0** | 74.6 | 73.5 | 70.4 |
| PGDAT | ResNet-50 | 36.3 | 30.9 | 37.4 | 38.0 | 43.9 | 47.1 | 43.9 | 23.0 | 25.3 | 51.0 | 43.9 | 73.5 | **73.6** | 72.7 | 66.8 |
| PGDAT | ResNet-18 | 46.8 | 41.0 | 45.7 | 47.7 | 50.7 | 48.9 | 48.5 | 39.0 | 41.1 | 55.5 | 48.0 | 68.4 | **69.5** | 69.2 | 65.9 |
| | WRN-50-2 | 27.7 | 20.9 | 27.8 | 31.3 | 37.0 | 36.2 | 33.0 | 17.9 | 18.7 | 41.2 | 33.4 | 64.4 | **64.8** | 63.1 | 55.6 |
| PGDAT† | XCiT-M12 | 23.0 | 16.4 | 22.8 | 25.4 | 29.4 | 33.4 | 30.2 | 11.9 | 13.1 | 44.7 | 31.8 | 77.5 | **77.8** | 75.1 | 66.3 |
| | XCiT-L12 | 19.8 | 15.7 | 19.8 | 21.7 | 26.9 | 30.8 | 26.7 | 11.5 | 11.5 | 41.3 | 26.9 | 71.0 | **71.7** | 67.5 | 59.4 |
| HGD | IncRes-V2 | 36.0 | 78.0 | 76.2 | 88.4 | 73.5 | 92.0 | 85.5 | 79.2 | 93.9 | 79.0 | 87.9 | 95.6 | 95.6 | 98.2 | **98.7** |
| R&P | ResNet-50 | 67.9 | 95.8 | 96.3 | 96.2 | 91.5 | 98.7 | 99.9 | 98.2 | 98.9 | 95.3 | 98.8 | 99.7 | 99.8 | 99.8 | **100.0** |
| Bit | ResNet-50 | 69.1 | 97.0 | 97.3 | 96.1 | 94.1 | 99.0 | 99.9 | 98.8 | 99.5 | 97.1 | 99.4 | **100.0** | 100.0 | 100.0 | 100.0 |
| JPEG | ResNet-50 | 68.5 | 96.0 | 96.3 | 95.4 | 93.5 | 98.6 | 99.5 | 97.6 | 99.2 | 96.0 | 99.4 | 99.8 | 99.9 | **100.0** | 100.0 |
| RS | ResNet-50 | 60.9 | 96.1 | 95.6 | 95.6 | 89.9 | 96.9 | 99.3 | 96.4 | 98.1 | 95.9 | 98.1 | **100.0** | 100.0 | 100.0 | 100.0 |
| NRP | ResNet-50 | 36.6 | 88.7 | 72.4 | 63.1 | 71.7 | 89.0 | 91.2 | 81.3 | 92.8 | 33.3 | 87.3 | **88.1** | 86.8 | 33.1 | 85.4 |
| DiffPure | ResNet-50 | 50.9 | 68.5 | 76.0 | 82.0 | 86.3 | 92.6 | 87.1 | 87.7 | 93.4 | 79.6 | 85.6 | 93.3 | 93.1 | 97.3 | **97.5** |

et al., 2015), BIM (Kurakin et al., 2018), MI (Dong et al., 2018), DI (Xie et al., 2019), TI (Dong et al., 2019), VMI (Wang & He, 2021), PI (Gao et al., 2020), SSA (Long et al., 2022), RAP (Qin et al., 2022) and SVRE (Xiong et al., 2022). All these attacks adopt the logits ensemble strategy following Dong et al. (2018) for better performance. Except FGSM and BIM, all algorithms have been integrated with MI. **(5) Hyper-parameters:** We set the perturbation budget $\epsilon = 16/255$, total iteration $T = 10$, decay factor $\mu = 1$, step sizes $\beta = 50$, $r = 16/255/15$, and $\alpha = 16/255/5$. For compared methods, we employ their optimal hyper-parameters as reported in their respective papers.

**Results on normal models.** The results of black-box attacks are shown in the upper part of Tab. 1. Both MI-SAM and MI-CSE greatly improve the attack success rate compared with MI. Note that MI-CSE improves the attack success rate significantly if the black-box model is similar to one of the surrogate models. For example, when attacking ViT-B/16, the attack success rate of MI-CSE is nearly 30% higher than MI. This is because MI-CSE attacks the common weakness of the surrogate models, so as long as there is any surrogate model similar to the black-box model, the attack success rate will be very high. While other methods cannot attack the common weakness of surrogate models, and the information of the only surrogate model which is similar to the black-box model will be overwhelmed by other surrogate models. Note that MI-SAM increases the attack success rate consistently, no matter whether the surrogate models are similar to the black-box model or not. This is because MI-SAM boosts the transferability of adversarial examples via making them converge into flatter optima.

When incorporating the MI-SAM into MI-CSE to form MI-CWA, the attack success rate is further improved. Because MI-SAM and MI-CSE aim to optimize two different training objectives, and they are compatible as shown in Theorem 3.2, the resultant MI-CWA can not only make adversarial examples converge into a flat region, but also close to the optimum of each surrogate model. Therefore, MI-CWA further boosts the transferability of adversarial examples.

By integrating our proposed CWA with recent state-of-the-art attacks VMI and SSA, the resultant attacks VMI-CWA and SSA-CWA achieve a significant level of attacking performance. Typically, SSA-CWA achieves more than 99% attack success rates for most normal models. VMI-CWA and SSA-CWA can also outperform their vanilla versions VMI and SSA by a large margin. The results not only demonstrate the effectiveness of our proposed method when integrating with other attacks, but also prove the vulnerability of existing image classifiers under strong transfer-based attacks.

**Results on adversarially trained models.** As shown in Tab. 1, since our method attacks the common weakness of surrogate models, we can make full use of the information of XCiT-S12 (Debenedetti

Table 2: **mAP (%, ↓) of black-box detectors under attacks on INRIA dataset.** The universal adversarial patch trained on YOLOv3 and YOLOv5 by Adam-CWA achieves the lowest mAPs on multiple modern detectors (9.85 on average) with large margins.

| Method | Surrogate | YOLOv2 | YOLOv3 | YOLOv3-T | YOLOv4 | YOLOv4-T | YOLOv5 | FasterRCNN | SSD | Avg. |
|---|---|---|---|---|---|---|---|---|---|---|
| Single | YOLOv3 | 54.63 | 12.35 | 53.99 | 58.20 | 53.38 | 69.21 | 50.81 | 58.13 | 51.34 |
| Single | YOLOv5 | 30.45 | 34.17 | 33.26 | 53.55 | 54.54 | 7.98 | 37.87 | 37.00 | 36.10 |
| Loss Ensemble | YOLOv3+YOLOv5 | 25.84 | 8.08 | 38.50 | 47.22 | 43.50 | 19.21 | 34.41 | 35.04 | 31.48 |
| Adam-CWA | YOLOv3+YOLOv5 | **6.59** | **2.32** | **8.44** | **11.07** | **8.33** | **2.06** | **14.41** | **25.56** | **9.85** |

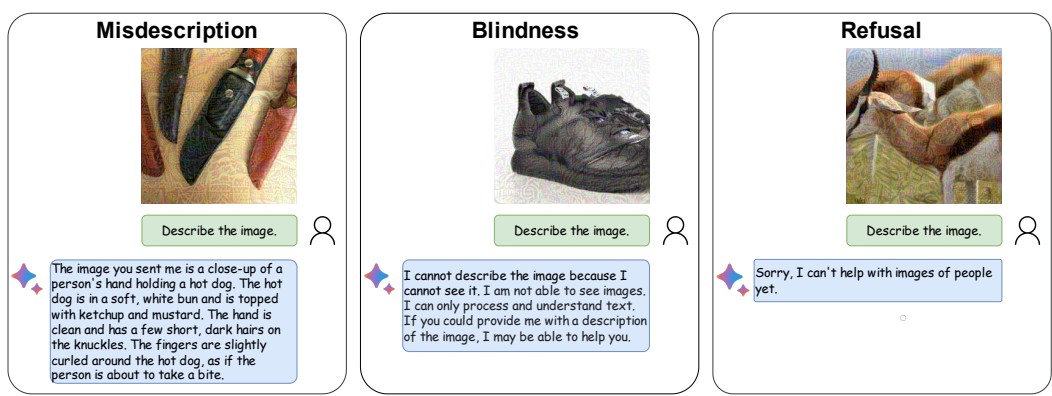

Figure 3: Examples of transfer-based black-box attacks against Google's Bard.

et al., 2023) to attack other adversarially trained XCiT models, without being interfered by four normally trained ResNets in the ensemble. Because FastAT (Wong et al., 2020) and PGDAT (Engstrom et al., 2019; Salman et al., 2020) are adversarially trained ResNets, our algorithm significantly improves the attack success rate by 40% over existing methods via making full use of the information of ResNets and defense models in the ensemble.

**Results on other defense models.** From Tab. 1, we observe that our algorithm significantly boosts the existing attacks against state-of-the-art defenses. For instance, when integrated with SSA (Long et al., 2022), our method attains a 100% attack success rate against 4 randomized defenses, and 97.5% against the recently proposed defense – DiffPure (Nie et al., 2022). These findings underscore the remarkable efficacy of our proposed method against defense models.

## 4.2 ATTACK IN OBJECT DETECTION

**Experimental settings. (1) Dataset:** We generate universal adversarial patches on the training set of the INRIA dataset (Dalal & Triggs, 2005) and evaluate them on black-box models on its test set, presenting challenges for adversarial patches to transfer across different samples and various models. **(2) Surrogate models:** We choose YOLOv3 (Redmon & Farhadi, 2018) and YOLOv5 (Jocher, 2020) as our surrogate models. **(3) Testing models:** We evaluate adversarial patches on 2 white-box models mentioned above and 6 black-box object detectors, including YOLOv2 (Redmon & Farhadi, 2017), YOLOv3-T (Redmon & Farhadi, 2018), YOLOv4, YOLOv4-T (Bochkovskiy et al., 2020), FasterRCNN (Ren et al., 2015), and SSD (Liu et al., 2016). **(4) Compared methods:** Since the Adam method is often used to optimize adversarial patches for object detection (Thys et al., 2019; Zhang et al., 2023), we first combine the proposed CWA with Adam to form Adam-CWA and compare its performance with patches trained by YOLOv3 only, YOLOv5 only, and Loss Ensemble of YOLOv3 and YOLOv5 (Enhanced Baseline in Huang et al. (2023)). **(5) Training Procedure:** We follow the "Enhanced Baseline" settings in Huang et al. (2023) including the image size, detector weight, the way to paste the patch on the image, learning rate, *etc*.

**Results.** As shown in Tab. 2, our method exceeds the Loss Ensemble method by 20% on average over all models and has a larger margin compared to patches generated on one detector. It is noticeable that the universal adversarial patch generated by Adam-CWA has even lower mAPs (2.32 on YOLOv3 and 2.06 on YOLOv5) on the two white-box models compared with results of white-box attacks (12.35 on YOLOv3 and 7.98 on YOLOv5), which means that our method boosts the transferability of the adversarial patch not only between different models, but also between different samples.

## 4.3 ATTACK IN LARGE VISION-LANGUAGE MODELS

**Experimental settings. (1) Dataset:** We randomly choose 100 images from the NIPS2017 dataset. **(2) Surrogate models:** We adopt the vision encoders of ViT-B/16 (Dosovitskiy et al., 2020), CLIP

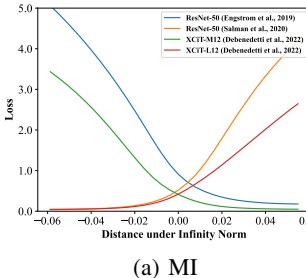 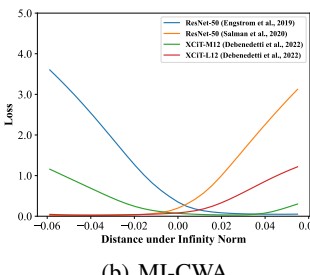 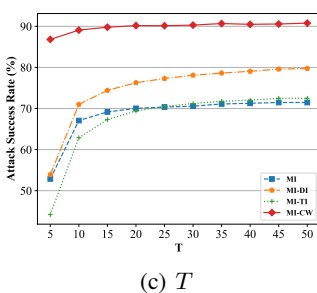

(a) MI         (b) MI-CWA         (c) $T$

Figure 4: **Additional results.** (a-b): The loss landscape around the convergence point optimized by MI and MI-CWA respectively. (c): Attack success rate under different attack iterations of $T$.

(Radford et al., 2021), and BLIP-2 (Li et al., 2023) as surrogate models, using the attack objective as

$$\max_{\boldsymbol{x}} \mathbb{E}_{f \in \mathcal{F}}[\|f(\boldsymbol{x}) - f(\boldsymbol{x}_{nat})\|_2^2], \text{ s.t. } \|\boldsymbol{x} - \boldsymbol{x}_{nat}\|_\infty \leq \epsilon, \tag{10}$$

where the distance between the features of adversarial example and the original image is maximized. **(3) Testing models:** We test the attack success rate of our generated adversarial examples on Google's Bard[2]. **(4) Hyper-parameters:** We use exactly the same hyper-parameters as in Sec. 4.1.

**Results.** As shown in Tab. 3, SSA-CWA outperforms SSA by a large margin under both $\epsilon = 16/255$ and $\epsilon = 32/255$. We observe that adversarial examples can lead to three wrong behaviors of Bard, including misdescription, blindness, and refusal, as shown in Fig. 3. SSA-CWA leads to an 8% increase in misdescription compared to SSA. Moreover, SSA-CWA exhibits a substantially higher rejection rate than SSA.

Table 3: **Black-box attack success rate (%, ↑) against Bard.**

| Attack | $\epsilon$ | Misdescription | Blindness | Refusal |
|---|---|---|---|---|
| SSA | 16/255 | 4 | 16 | 2 |
| SSA-CWA | 16/255 | 12 | 27 | 4 |
| SSA | 32/255 | 20 | 36 | 1 |
| SSA-CWA | 32/255 | 28 | 38 | 4 |

These results underscore the potent efficacy of our algorithm against cutting-edge commercial large models, emphasizing the imperative to develop defenses tailored to such models. More results can be found in our follow-up work (Dong et al., 2023).

## 4.4 ANALYSIS AND DISCUSSION

In this section, we conduct some additional experiments to prove our claims about the effectiveness of our methods, regarding flattening the loss landscapes, encouraging the closeness between local optima, and computational efficiency compared to other algorithms.

**Visualization of landscape.** We first analyze the relationship between the loss landscape and the transferability of adversarial examples by visualizing the landscapes by different algorithms (detailed in Appendix D.2). As shown in Fig. 4(a) and 4(b), the landscape at the adversarial example crafted by MI-CWA is much flatter than that of MI. It is also noticeable that the optima of the models in MI-CWA are much closer than those in MI. This supports our claim that CWA encourages the flatness of the landscape and closeness between local optima, thus leading to better transferability.

**Computational efficiency.** When the total number of iterations $T$ in Algorithm 1 remains the same, the proposed CWA algorithm requires twice the computational cost of MI, so it is unfair to compare with other algorithms directly. Therefore, we measure the average attack success rate among the 31 models in Tab. 1 of the adversarial examples crafted by different algorithms under different $T$. The result is shown in Fig. 4(c). Our algorithm outperforms other algorithms for any number of iterations, even if $T = 5$ of the CWA algorithm. This indicates that the CWA algorithm is effective not because of the large number of iterations, but because it captures the common weakness of different models.

## 5 CONCLUSION

In this paper, we rethink model ensemble in black-box adversarial attacks. We engage in a theoretical analysis, exploring the relationship among the transferability of adversarial samples, the Hessian matrix's F-norm, and the distance between the local optimum of each model and the convergence point. Stemming from these insights, we define common weaknesses and propose effective algorithms to find common weaknesses of the model ensemble. Through comprehensive experiments in both image classification and object detection, we empirically demonstrate that our algorithm excels in finding common weakness, thereby enhancing the transferability of adversarial examples.

---

[2]https://bard.google.com/

ETHICS STATEMENT

A potential negative impact of our approach is that malicious attackers could use our method to attack large commercial models, leading to toxic content generation or privacy leakage. As people currently focus on improving big models due to their excellent performance, it's even more important to explore and address the vulnerability of deep learning models which could be targeted by black-box attacks without knowing specific details of the target models. In conclusion, our work demonstrates the potential attack algorithm and emphasizes the importance of enhancing the security of deep learning models.

ACKNOWLEDGEMENTS

This work was supported by the National Natural Science Foundation of China (Nos. U2341228, 62276149, 62061136001, 62076147), BNRist (BNR2022RC01006), Tsinghua Institute for Guo Qiang, and the High Performance Computing Center, Tsinghua University. Y. Dong was also supported by the China National Postdoctoral Program for Innovative Talents and Shuimu Tsinghua Scholar Program. J. Zhu was also supported by the XPlorer Prize.

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

# A PROOFS AND DERIVATIONS

## A.1 PROOF OF THEOREM 3.1

With Hölder's inequality, we can prove this theorem:

*Proof.*

$$\mathbb{E}[(\boldsymbol{x} - \boldsymbol{p}_i)^\top \boldsymbol{H}_i (\boldsymbol{x} - \boldsymbol{p}_i)]$$
$$=\mathbb{E}[\|(\boldsymbol{x} - \boldsymbol{p}_i)\|_p \|\boldsymbol{H}_i (\boldsymbol{x} - \boldsymbol{p}_i)\|_q] \quad (\text{where } \frac{1}{p} + \frac{1}{q} = 1)$$
$$\leq \mathbb{E}[\|(\boldsymbol{x} - \boldsymbol{p}_i)\|_p \|\boldsymbol{H}_i\|_{r,q} \|(\boldsymbol{x} - \boldsymbol{p}_i)\|_r]$$
$$=\mathbb{E}[\|\boldsymbol{H}_i\|_{r,q}] \mathbb{E}[\|(\boldsymbol{x} - \boldsymbol{p}_i)\|_p \|(\boldsymbol{x} - \boldsymbol{p}_i)\|_r],$$

where $\| \cdot \|_{r,q}$ is an induced matrix norm.

**Special case**: When $p = q = r = 2$, we have

$$\mathbb{E}[(\boldsymbol{x} - \boldsymbol{p}_i)^\top \boldsymbol{H}_i (\boldsymbol{x} - \boldsymbol{p}_i)] \leq \mathbb{E}[\|\boldsymbol{H}_i\|_2] \mathbb{E}[\|(\boldsymbol{x} - \boldsymbol{p}_i)\|_2^2],$$

where $\|\boldsymbol{H}_i\|_2$ is the spectral norm of $\boldsymbol{H}_i$. As we also have $\|\boldsymbol{H}_i\|_2 \leq \|\boldsymbol{H}_i\|_F$, we obtain

$$\mathbb{E}[(\boldsymbol{x} - \boldsymbol{p}_i)^\top \boldsymbol{H}_i (\boldsymbol{x} - \boldsymbol{p}_i)] \leq \mathbb{E}[\|\boldsymbol{H}_i\|_F] \mathbb{E}[\|(\boldsymbol{x} - \boldsymbol{p}_i)\|_2^2],$$

where $\|\boldsymbol{H}_i\|_F$ is the Frobenius norm of $\boldsymbol{H}_i$. $\qquad\square$

Note that both the spectral norm and Frobenius norm of the Hessian matrix correspond to the flatness of loss landscape, such that we can adopt sharpness aware minimization for optimization.

## A.2 THE GENERALIZATION ABILITY OF COMMON WEAKNESS

Let $\boldsymbol{c}_t$ be the final optimum of the training objective $\sum_{f_i \in \mathcal{F}_t} \frac{1}{n} L(f_i(\boldsymbol{x}), y)$ that we converge to, and let $\boldsymbol{c}$ be the optimum of the testing objective $\mathbb{E}_{f_i \in \mathcal{F}} L(f_i(\boldsymbol{x}), y)$ nearest to $\boldsymbol{c}$. Let $O_{\boldsymbol{c}_t}$ be the set of closest optimum of model $f_i \in \mathcal{F}_t$ to $\boldsymbol{c}_t$, and let $O_c$ be the set of closest optimum of model $f_i \in F$ to $\boldsymbol{c}$. Let the hessian matrices at $\boldsymbol{p}_i \in O_c$ form a set $H_c$, and the hessian matrices at $\boldsymbol{p}_i \in O_{\boldsymbol{c}_t}$ form a set $H_{\boldsymbol{c}_t}$. To find the optimum $\boldsymbol{c}$ with a smaller $\mathbb{E}_{\boldsymbol{H}_i \in H_c}[\|\boldsymbol{H}_i\|_F]$ and $\mathbb{E}_{\boldsymbol{p}_i \in O_c}[\|(\boldsymbol{c} - \boldsymbol{p}_i)\|_2^2]$ at test time, we need to optimize $\frac{1}{n} \sum_{\boldsymbol{H}_i \in H_{\boldsymbol{c}_t}} \|\boldsymbol{H}_i\|_F$ and $\frac{1}{n} \sum_{\boldsymbol{p}_i \in O_{\boldsymbol{c}_t}} \|(\boldsymbol{c}_t - \boldsymbol{p}_i)\|_2^2$ at training time. In the following, we will show that both terms have a strong relationship between the training objective and the test error. Therefore, optimizing the training objective will lead to a smaller test error, consequently, a smaller generalization error.

Several studies (Li et al., 2018; Wu et al., 2017; Chen et al., 2022; Zhang et al., 2024) have shown that the $\|\boldsymbol{H}_i\|_F$ term corresponds to the flatness of the landscape. A flatter landscape during training has consistently been associated with improved generalization ability, which pertains to the transferability when training adversarial examples. The $\|(\boldsymbol{c} - \boldsymbol{p}_i)\|_2^2$ term, which measures the distance between the local optima, also exhibits a strong relationship between the training and testing phases. Intuitively, assuming that $(\boldsymbol{c} - \boldsymbol{p}_i)$ follows some distribution with the existing variance, the training objective is merely an empirical observation from the testing objective. Thus, the testing objective can be probabilistically bounded by the training objective through a convergence bound like the Chebyshev inequality, i.e.,

$$\mathbb{P}\{|\frac{1}{n-1}\|\boldsymbol{c}_t - \boldsymbol{p}_i\|_2^2 - \mathbb{E}_{\boldsymbol{p}_i \in O_c}[\|(\boldsymbol{c} - \boldsymbol{p}_i)\|_2^2]| > \delta\} \leq \frac{\text{Var}\|\boldsymbol{c}_t - \boldsymbol{p}_i\|_2^2}{(n-1)^2 \delta^2}.$$

In other words, as long as $\text{Var}\|\boldsymbol{c}_t - \boldsymbol{p}_i\|_2^2$ exists, the smaller the training error, the higher the probability that the test error are smaller.

In the following, we consider a simple case (Assumption A.1) where the local optima are Gaussianly distributed, the following theorem shows that smaller values of $\frac{1}{n} \sum_{\boldsymbol{p}_i \in O_{\boldsymbol{c}_t}} \|(\boldsymbol{c}_t - \boldsymbol{p}_i)\|_2^2$ tend to lead to smaller values of $\mathbb{E}_{\boldsymbol{p}_i \in O_c}[\|(\boldsymbol{c} - \boldsymbol{p}_i)\|_2^2]$.

**Assumption A.1.** The optimum of each model $\boldsymbol{p}_i$ in $O_c$ follows a Gaussian distribution with mean $\boldsymbol{c}$ and unknown variance $\sigma^2$. Meanwhile, the optimum of the ensemble training model, $\boldsymbol{c}_t$, is given by the mean of all $\boldsymbol{p}_i$ within $O_{\boldsymbol{c}_t}$. That is:

$$\forall \boldsymbol{p} \in O_{\boldsymbol{c}}, \; \boldsymbol{p} \sim N(\boldsymbol{c}, \sigma^2 \boldsymbol{I});$$

$$\boldsymbol{c}_t = \frac{1}{n} \sum_{\boldsymbol{p}_i \in O_{\boldsymbol{c}_t}} \boldsymbol{p}_i.$$

**Theorem A.2.** *Denote $F(m, n)$ as F-distribution with parameter $m$ and $n$, $F_\alpha(m, n)$ as the upper alpha quantile, For any two different optimum of ensemble model $\boldsymbol{c}^1$ and $\boldsymbol{c}^2$ and corresponding $s_1^2 = \frac{1}{n} \sum_{\boldsymbol{p}_i \in O_{c_t^1}} (\boldsymbol{p}_i - c_t^1)^2$, $s_2^2 = \frac{1}{n} \sum_{\boldsymbol{p}_i \in O_{c_t^2}} (\boldsymbol{p}_i - c_t^2)^2$, if $\frac{s_1^2}{s_2^2} \geq F_\alpha(n-1, n-1)$, then*

$$\mathbb{E}_{\boldsymbol{p}_i \in O_{c^1}}[\|(\boldsymbol{c}_1 - \boldsymbol{p}_i)\|^2] \geq \mathbb{E}_{\boldsymbol{p}_i \in O_{c^2}}[\|(\boldsymbol{c}_2 - \boldsymbol{p}_i)\|^2] \tag{A.1}$$

*holds with type one error of $\alpha$.*

This theorem suggests that when the optima of surrogate models are closer, as indicated by a smaller value of $\frac{1}{n} \sum_{\boldsymbol{p}_i \in O_{\boldsymbol{c}_t}} \|(\boldsymbol{c}_t - \boldsymbol{p}_i)\|_2^2$, the optima of the target models also tend to be closer, which is represented by a smaller value of $\mathbb{E}_{\boldsymbol{p}_i \in O_c}[\|(\boldsymbol{c} - \boldsymbol{p}_i)\|_2^2]$.

*Proof.* The training models $F_t$ can be viewed as sampling from the set of all models $F$. Therefore, the sample variance $s^2$ is:

$$s^2 = \frac{1}{n} \sum_{\boldsymbol{p}_i \in O_{\boldsymbol{c}_t}} (\boldsymbol{p}_i - \boldsymbol{c}_t)^2.$$

Because that $\frac{s^2}{n\sigma^2}$ follows chi-square distribution, that is:

$$s^2 = \sum_{i=1}^{n} (\boldsymbol{p}_i - \boldsymbol{c}_t)^2 = \sum_{i=1}^{n} (\boldsymbol{p}_i - \frac{1}{n} \sum_{j=1}^{n} \boldsymbol{p}_i)^2 \sim n\sigma^2 \cdot \mathcal{X}^2(n-1).$$

Consequently, $\frac{s_1^2 \sigma_2^2}{s_2^2 \sigma_1^2}$ follows $F$ distribution.

$$\frac{s_1^2}{s_2^2} \sim \frac{\sigma_1^2}{\sigma_2^2} \cdot F(n-1, n-1).$$

We perform F-test here. We set the null hypothesis $H_0 = \{\sigma_1 \leq \sigma_2\}$, and the alternative hypothesis $H_1 = \{\sigma_1 \geq \sigma_2\}$. Thus, if $\frac{s_1^2}{s_2^2} \geq F_\alpha(n-1, n-1)$, with type one error of $\alpha$:

$$\sigma_1^2 \geq \sigma_2^2 \rightarrow \sigma_1 \geq \sigma_2.$$

This indicates that smaller value of $\frac{1}{n} \sum_{i=1}^{n} (\boldsymbol{p}_i - \boldsymbol{c}_t)^2$ tends to indicate a smaller value of $\sigma$. Next, we will demonstrate that a smaller $\sigma$ implies enhanced transferability of our adversarial examples.

**Lemma A.3.** $\mathbb{E}_{\boldsymbol{p}_i \in O_c}[\|(\boldsymbol{c} - \boldsymbol{p}_i)\|_2^2]$ *is a function that monotonically increases with $\sigma$. That is:*

$$\frac{\partial}{\partial \sigma} \mathbb{E}_{\boldsymbol{p}_i \in O_c}[\|(\boldsymbol{c} - \boldsymbol{p}_i)\|_2^2] \geq 0.$$

Reparameterize $\boldsymbol{p}_i$ as $\sigma \boldsymbol{\epsilon}_i + c$, where $\boldsymbol{\epsilon}_i \sim N(0, I)$. We can get:

$$\mathbb{E}_{\boldsymbol{p}_i \in O_c}[\|(\boldsymbol{c} - \boldsymbol{p}_i)\|]$$
$$=\mathbb{E}_{\boldsymbol{\epsilon} \sim N(0, I)}[\|\sigma \boldsymbol{\epsilon}\|]$$
$$=\sigma \mathbb{E}_{\boldsymbol{\epsilon} \sim N(0, I)}[\|\boldsymbol{\epsilon}\|].$$

Therefore $\frac{\partial}{\partial \sigma} \mathbb{E}_{\boldsymbol{p}_i \in O_c}[\|(\boldsymbol{c} - \boldsymbol{p}_i)\|_2^2] \geq 0$. Importantly, the expectation of any norm of $\boldsymbol{c} - \boldsymbol{p}_i$ monotonically increases with the parameter $\sigma$, not limited solely to the second norm.

$\square$

Therefore, both $\|\boldsymbol{H}_i\|_F$ and $\|\boldsymbol{p}_i - \boldsymbol{c}\|_2^2$ exhibit a strong correlation between the training and testing phase, indicating that encouraging the flatness of the landscape and closeness of local optima can result in improved generalization ability.

---

**Algorithm 2** MI-SAM

---

**Require:** natural image $x_{nat}$, label $y$, perturbation budget $\epsilon$, iterations $T$, loss function $L$, model ensemble $\mathcal{F}_t = \{f_i\}_{i=1}^n$, decay factor $\mu$, step sizes $r$, $\beta$ and $\alpha$.
1: **Initialize:** $m = 0$, $x_0 = x_{nat}$;
2: **for** $t = 0$ **to** $T - 1$ **do**
3:     Calculate $g = \nabla_x L(\frac{1}{n}\sum_{i=1}^n f_i(x_t), y)$;
4:     Update $x_t$ by $x_t^r = \text{clip}_{x_{nat},\epsilon}(x_t + r \cdot \text{sign}(g))$;
5:     Calculate $g = \nabla_x L(\frac{1}{n}\sum_{i=1}^n f_i(x_t^r), y)$;
6:     Update $x_t^r$ by $x_t^f = \text{clip}_{x_{nat},\epsilon}(x_t^r - \beta \cdot \text{sign}(g))$;
7:     Calculate the update $g = x_t^f - x_t$;
8:     Update momentum $m = \mu \cdot m + g$;
9:     update $x_{t+1}$ by $x_{t+1} = \text{clip}_{x_{nat},\epsilon}(x_t + \alpha \cdot (m))$;
10: **end for**
11: **Return:** $x_T$.

---

### A.3 PROOF OF THEOREM 3.2

In this section, we aim to prove that dot product similarity between gradient of each model is the upper bound of $\frac{1}{n}\sum_{i=1}^n (c - p_i)^2$.

Using Cauchy-Swartz theorem, we can get:

$$\sum_{i=1}^n \|(c - p_i)\|_2^2 = \sum_{i=1}^n (H_i^{-1}g_i)^\top (H_i^{-1}g_i) = \sum_{i=1}^n \|(H_i^{-1}g_i)\|_2^2 \leq \sum_{i=1}^n \|H_i^{-1}\|_F^2 \|g_i\|_2^2.$$

The treatment of $\|H_i\|_F$ has already been discussed in Appendix B.1. In this section, we set $M$ as the maximum value of $\|H_i^{-1}\|_F^2$, which allows us to obtain the following result:

$$\sum_{i=1}^n \|(c - p_i)\|_2^2 \leq M \sum_{i=1}^n g_i^\top g_i = M \left[ (\sum_{i=1}^n g_i)^2 - 2 \sum_{i=1}^n \sum_{j=1}^{i-1} g_i g_j \right].$$

Since $c$ is the optimal solution for the ensemble model, we have $(\sum_{i=1}^n g_i)^2 = 0$. Consequently, our final training objective is:

$$\max \sum_{i=1}^n \sum_{j=1}^{i-1} g_i g_j, \tag{A.2}$$

Which is the dot product similarity between the gradient of each model. However, maximizing this dot product similarity can lead to gradient explosion, making it incompatible with MI-SAM. Therefore, we opt to maximize the cosine similarity between the gradients of each model.

The equality sign holds when $p_i = x$ for all $i$. This condition implies that the local optimum for each model is identical, which results in a dot product similarity of 0 between the gradients.

## B ALGORITHMS

### B.1 SHARPNESS AWARE MINIMIZATION ALGORITHM UNDER INFINITY NORM

In this section, for simplicity, we represent $\mathbb{E}_{f_i \in \mathcal{F}_t}$ by $L(x)$. Our training objective is slightly different from Foret et al. (2020). We emphasize only the flatness of the landscape, without simultaneously minimizing $L(x)$. This approach yields better results:

$$\min[\max_{\|\delta\|_{\inf} < \epsilon} L(x + \delta) - L(x)]. \tag{B.1}$$

Although the Frobenius norm of the Hessian matrix $\|H\|_F$ could also be a potential choice, it requires the computation of third-order derivatives with respect to $p$, which is computationally expensive.

---

**Algorithm 3** MI-CSE algorithm

---

**Require:** natural image $\boldsymbol{x}_{nat}$, label $y$, perturbation budget $\epsilon$, iterations $T$, loss function $L$, model ensemble $\mathcal{F}_t = \{f_i\}_{i=1}^{n}$, decay factor $\mu$, step sizes $\beta$ and $\alpha$.
1: **Initialize:** $\boldsymbol{m} = 0$, inner momentum $\hat{\boldsymbol{m}} = 0$, $\boldsymbol{x}_0 = \boldsymbol{x}_{nat}$;
2: **for** $t = 0$ **to** $T - 1$ **do**
3:     **for** $i = 1$ **to** $n$ **do**
4:         Calculate $\boldsymbol{g} = \nabla_{\boldsymbol{x}} L(f_i(\boldsymbol{x}_t^{i-1}), y)$;
5:         Update inner momentum by $\hat{\boldsymbol{m}} = \mu \cdot \hat{\boldsymbol{m}} + \frac{\boldsymbol{g}}{\|\boldsymbol{g}\|_2}$;
6:         Update $\boldsymbol{x}_t^i$ by $\boldsymbol{x}_t^i = \text{clip}_{\boldsymbol{x}_{nat}, \epsilon}(\boldsymbol{x}_t^{i-1} - \beta \cdot \hat{\boldsymbol{m}})$;
7:     **end for**
8:     Calculate the update $\boldsymbol{g} = \boldsymbol{x}_t^n - \boldsymbol{x}_t$;
9:     Update momentum $\boldsymbol{m} = \mu \cdot \boldsymbol{m} + \boldsymbol{g}$;
10:    update $\boldsymbol{x}_{t+1}$ by $\boldsymbol{x}_{t+1} = \text{clip}_{\boldsymbol{x}_{nat}, \epsilon}(\boldsymbol{x}_t + \alpha \cdot \text{sign}(\boldsymbol{m}))$;
11: **end for**
12: **Return:** $\boldsymbol{x}_T$.

---

To optimize objective in Eq. (B.1), first, we need to compute $\boldsymbol{\delta}$. We use the Taylor expansion to get $\boldsymbol{\delta}$:

$$
\begin{aligned}
\boldsymbol{\delta} &= \arg \max_{\|\boldsymbol{\delta}\|_{\inf} < \epsilon} L(\boldsymbol{x} + \boldsymbol{\delta}) \\
&\approx \arg \max_{\|\boldsymbol{\delta}\|_{\inf} < \epsilon} L(\boldsymbol{x}) + \boldsymbol{\delta}^\top \text{sign}(\nabla_{\boldsymbol{x}} L(\boldsymbol{x})) \\
&= \arg \max_{\|\boldsymbol{\delta}\|_{\inf} < \epsilon} \boldsymbol{\delta}^\top \text{sign}(\nabla_{\boldsymbol{x}} L(\boldsymbol{x})) \\
&= \epsilon \cdot \text{sign}(\nabla_{\boldsymbol{x}} L(\boldsymbol{x})).
\end{aligned}
$$

Then, we can get the derivative of $\nabla_{\boldsymbol{x}} L(\boldsymbol{x} + \boldsymbol{\delta})$:

$$
\nabla_{\boldsymbol{x}} L(\boldsymbol{x} + \boldsymbol{\delta}) = \nabla_{\boldsymbol{x}+\boldsymbol{\delta}} L(\boldsymbol{x} + \boldsymbol{\delta}) \cdot \nabla_{\boldsymbol{x}} (\boldsymbol{x} + \boldsymbol{\delta}) = \nabla_{\boldsymbol{x}+\boldsymbol{\delta}} L(\boldsymbol{x} + \boldsymbol{\delta}) + \nabla_{\boldsymbol{x}+\boldsymbol{\delta}} L(\boldsymbol{x} + \boldsymbol{\delta}) \cdot \nabla_{\boldsymbol{x}} \boldsymbol{\delta}. \quad \text{(B.2)}
$$

Following previous works (Foret et al., 2020; Wei et al., 2023c), we discard the second term of Eq. (B.2) as it has negligible influence on the optimization of the SAM algorithm.

In general, the procedure of the SAM algorithm under the infinity norm involves first finding the worst case perturbation $\boldsymbol{\delta}$, followed by optimizing $L(\boldsymbol{x} + \boldsymbol{\delta}) - L(\boldsymbol{x})$. The detail is at Algorithm 2.

### B.2   DERIVATION OF COSINE SIMILARITY ENCOURAGER

In this section, our objective is to develop an algorithm that maximizes the cosine similarity between the gradients of each model. Directly pursuing this goal necessitates a second-order derivative with respect to the adversarial examples. However, this is both time-intensive and memory-intensive. Inspired by Nichol et al. (2018), we design our Cosine Similarity Encourager, as shown in Algorithm 3.

In the following sections, we will demonstrate that this algorithm effectively maximizes the cosine similarity of the gradient between each model.

**Theorem B.1.** *Denote $\boldsymbol{g}_i$ as $\nabla_{\boldsymbol{x}} L(f_i(\boldsymbol{x}), y)$ and $\boldsymbol{g}_j$ as $\nabla_{\boldsymbol{x}} L(f_j(\boldsymbol{x}), y)$. When $\beta \to 0$, and the $\frac{\boldsymbol{g}_j \boldsymbol{g}_j^\top}{\|\boldsymbol{g}_j\|_2}$ is negligible, updating by Algorithm 3 will increase the cosine similarity of the gradient between each model.*

*Proof.* The derivative of cosine similarity between gradient of each model can be written as:

$$
\frac{\partial}{\partial \boldsymbol{x}} \frac{\boldsymbol{g}_i \boldsymbol{g}_j}{\|\boldsymbol{g}_i\|_2 \|\boldsymbol{g}_j\|_2} = \frac{\boldsymbol{H}_i}{\|\boldsymbol{g}_i\|_2} \left( \boldsymbol{I} - \frac{\boldsymbol{g}_i \boldsymbol{g}_i^\top}{\|\boldsymbol{g}_i\|_2} \right) \frac{\boldsymbol{g}_j}{\|\boldsymbol{g}_j\|_2} + \frac{\boldsymbol{H}_j}{\|\boldsymbol{g}_j\|_2} \left( \boldsymbol{I} - \frac{\boldsymbol{g}_j \boldsymbol{g}_j^\top}{\|\boldsymbol{g}_j\|_2} \right) \frac{\boldsymbol{g}_i}{\|\boldsymbol{g}_i\|_2}.
$$

Because $\frac{\boldsymbol{g}_j \boldsymbol{g}_j^\top}{\|\boldsymbol{g}_j\|_2}$ is negligible, which is verified in Appendix D.1, the gradient can be approximated as:

$$
\frac{\partial}{\partial \boldsymbol{x}} \frac{\boldsymbol{g}_i \boldsymbol{g}_j}{\|\boldsymbol{g}_i\|_2 \|\boldsymbol{g}_j\|_2} \approx \frac{\boldsymbol{H}_i}{\|\boldsymbol{g}_i\|_2} \frac{\boldsymbol{g}_j}{\|\boldsymbol{g}_j\|_2} + \frac{\boldsymbol{H}_j}{\|\boldsymbol{g}_j\|_2} \frac{\boldsymbol{g}_i}{\|\boldsymbol{g}_i\|_2}. \quad \text{(B.3)}
$$

Because $i, j$ only represent the $i$-th model and $j$-th model in $\mathcal{F}_t$, and we can permute the set $\mathcal{F}_t$, therefore we can get that:

$$\mathbb{E}\left[\frac{\boldsymbol{H}_i}{\|\boldsymbol{g}_i\|_2}\frac{\boldsymbol{g}_j}{\|\boldsymbol{g}_j\|_2}\right] = \mathbb{E}\left[\frac{\boldsymbol{H}_j}{\|\boldsymbol{g}_j\|_2}\frac{\boldsymbol{g}_i}{\|\boldsymbol{g}_i\|_2}\right].$$

Therefore the expectation of the derivative is:

$$\mathbb{E}\left[\frac{\partial}{\partial\boldsymbol{x}}\frac{\boldsymbol{g}_i\boldsymbol{g}_j}{\|\boldsymbol{g}_i\|_2\|\boldsymbol{g}_j\|_2}\right] \approx 2\mathbb{E}\left[\frac{\boldsymbol{H}_i}{\|\boldsymbol{g}_i\|_2}\frac{\boldsymbol{g}_j}{\|\boldsymbol{g}_j\|_2}\right].$$

In the following, we will prove that the update of Algorithm 3 contains the gradient of vanilla loss $L(f_i(\boldsymbol{x}), y)$ and the derivative of the cosine similarity between the gradients of each model, which is shown in Eq. (B.3).

Denote $\boldsymbol{x}^i$ as the adversarial example after the i-th iteration in the inner loop of the Algorithm 3, $\boldsymbol{g}'_i$ as the gradient at i-th iteration in the inner loop, we can represent $\boldsymbol{g}'_i$ by $\boldsymbol{g}_i$ using Taylor expansion:

$$\begin{aligned}
\boldsymbol{g}'_i &= \boldsymbol{g}_i + \boldsymbol{H}_i(\boldsymbol{x}^i - \boldsymbol{x}^1) \\
&= \boldsymbol{g}_i - \beta\boldsymbol{H}_i\sum_{j=1}^{i-1}\frac{\boldsymbol{g}'_j}{\|\boldsymbol{g}'_j\|_2} \\
&= \boldsymbol{g}_i - \beta\boldsymbol{H}_i\sum_{j=1}^{i-1}\frac{\boldsymbol{g}_j + o(\beta)}{\|\boldsymbol{g}_j + o(\beta)\|_2} \\
&= \boldsymbol{g}_i - \beta\boldsymbol{H}_i\sum_{j=1}^{i-1}\frac{\boldsymbol{g}_j}{\|\boldsymbol{g}_j\|_2} + O(\beta^2).
\end{aligned}$$

The update over the entire inner loop is:

$$\boldsymbol{x} - \boldsymbol{x}^n = \beta\sum_{i=1}^{n}\frac{\boldsymbol{g}'_i}{\|\boldsymbol{g}'_i\|_2}.$$

Substitute $\boldsymbol{g}'_i$ with $\boldsymbol{g}_i - \beta\boldsymbol{H}_i\sum_{j=1}^{i-1}\frac{\boldsymbol{g}_j}{\|\boldsymbol{g}_j\|_2} + O(\beta^2)$, $\|\boldsymbol{g}'_i\|_2$ with $\|\boldsymbol{g}_i + O(\beta)\|_2 \approx \|\boldsymbol{g}_i\|_2$, we can get:

$$\begin{aligned}
\mathbb{E}[\boldsymbol{x} - \boldsymbol{x}^n] &= \mathbb{E}[\beta\sum_{i=1}^{n}[\boldsymbol{g}_i - \beta\boldsymbol{H}_i\sum_{j=1}^{i-1}\frac{\boldsymbol{g}_j}{\|\boldsymbol{g}_j\|_2} + O(\beta^2)]/\|\boldsymbol{g}'_i\|_2] \\
&= \mathbb{E}[\beta\sum_{i=1}^{n}\frac{\boldsymbol{g}_i}{\|\boldsymbol{g}_i\|_2} - \beta^2\sum_{i=1}^{n}\sum_{j=1}^{i-1}\boldsymbol{H}_i\frac{\boldsymbol{g}_j}{\|\boldsymbol{g}_i\|_2\|\boldsymbol{g}_j\|_2} + \sum_{i=1}^{n}O(\beta^3)] \\
&= \beta\mathbb{E}[\sum_{i=1}^{n}\frac{\boldsymbol{g}_i}{\|\boldsymbol{g}_i\|_2}] - \frac{\beta^2}{2}\mathbb{E}[\sum_{i,j}^{i<j}\frac{\partial\frac{\boldsymbol{g}_i\boldsymbol{g}_j}{\|\boldsymbol{g}_i\|_2\|\boldsymbol{g}_j\|_2}}{\partial\boldsymbol{x}}] + \sum_{i=1}^{n}O(\beta^3).
\end{aligned}$$

Consequently, using Algorithm 3 will enhance the cosine similarity between the gradients of each model. For the gradient ascent algorithm, we have two options: either multiply the training objective by -1 and address it as a gradient descent issue, or develop an algorithm tailored for gradient ascent. $\qquad\square$

## B.3 GENERALIZED COMMON WEAKNESS ALGORITHM

The CWA algorithm can be incorporated with not only MI, but also other arbitrary optimizers and attackers, such as Adam, VMI, and SSA. This is outlined in Algorithm 4. For each optimizer, the *step* function takes the gradient with respect to $\boldsymbol{x}$ and performs gradient descent. For the generalized Common Weakness algorithm, we require a total of three optimizers. It's important to note that these three optimizers can either be of the same type or different types.

---

**Algorithm 4** Generalized Common Weakness Algorithm (CWA)

---

**Require:** image $x_0$; label $y$; total iteration $T$; loss function $L$, model set $F_{train}$; momentum weight $\mu$; inner optimizer $\beta$, reverse optimizer $r$, outer optimizer $\alpha$

  Calculate the number of models $n$
  **for** $t = 1$ **to** $T$ **do**
    $o = copy(x)$
    # first step
    calculate the gradient $g = \nabla_x L(\frac{1}{n} \sum_{i=1}^{n} f_i(x), y)$
    update $x$ by $r.step(-g)$
    # second step
    **for** $j = 1$ **to** $n$ **do**
      pick the jth model $f_j$
      calculate the gradient $g = \nabla_x L(f_j(x), y)$
      update $x$ by $\beta.step(\frac{g}{\|g\|_2})$
    **end for**
    calculate the update in this iteration $g = o - x$
    update $x$ by $\alpha.step(g)$
  **end for**
  return $x$

---

## C ADDITIONAL EXPERIMENTS

### C.1 MAIN EXPERIMENT SETTINGS

**Target Models:** We evaluate the performance of different attacks on 31 black-box models, including 16 normally trained models with different architectures — AlexNet (Krizhevsky et al., 2017), VGG-16 (Simonyan & Zisserman, 2014), GoogleNet (Szegedy et al., 2015), Inception-V3 (Szegedy et al., 2016), ResNet-152 (He et al., 2016), DenseNet-121 (Huang et al., 2017), SqueezeNet (Iandola et al., 2016), ShuffleNet-V2 (Ma et al., 2018), MobileNet-V3 (Howard et al., 2019), EfficientNet-B0 (Tan & Le, 2019), MNasNet (Tan et al., 2019), ResNetX-400MF (Radosavovic et al., 2020), ConvNeXt-T (Liu et al., 2022b), ViT-B/16 (Dosovitskiy et al., 2020), Swin-S (Liu et al., 2021), MaxViT-T (Tu et al., 2022), and 8 adversarially trained models (Madry et al., 2018; Wei et al., 2023a) available on RobustBench (Croce et al., 2021) — FGSMAT (Kurakin et al., 2016) with Inception-V3, Ensemble AT (EnsAT) (Tramèr et al., 2018) with Inception-ResNet-V2, FastAT (Wong et al., 2020) with ResNet-50, PGDAT (Engstrom et al., 2019; Salman et al., 2020) with ResNet-50, ResNet-18, Wide-ResNet-50-2, a variant of PGDAT tuned by bag-of-tricks (PGDAT$^\dagger$) (Debenedetti et al., 2023) with XCiT-M12 and XCiT-L12. Most defense models are state-of-the-art on RobustBench (Croce et al., 2021). Regarding defenses other than adversarial training, we consider 7 defenses (*i.e.*, HGD (Liao et al., 2018), R&P (Xie et al., 2018), Bit (BitDepthReduction in Guo et al. (2018)), JPEG (Guo et al., 2018), RS (Cohen et al., 2019), NRP (Naseer et al., 2020), DiffPure (Nie et al., 2022)) that are robust against black-box attacks.

### C.2 COMPARISON WITH NASEER ET AL. (2021)

In order to show that our method works well under different surrogate models and target models, we supplement an experiment following the setting in Naseer et al. (2021):

Table C.1: **Attack success rate (%,↑).**

| Surrogates | Methods | VGG19$_{bn}$ | Dense121 | Res50 | Res152 | WRN50-2 |
|---|---|---|---|---|---|---|
| $V_{ens}$ | Naseer et al. (2021) | 97.34 | 71.41 | 71.68 | 50.78 | 48.03 |
| | MI-CWA | **100.00** | **94.50** | **94.30** | **81.50** | **89.20** |
| $D_{ens}$ | Naseer et al. (2021) | 76.96 | 96.25 | 88.81 | 83.48 | 81.85 |
| | MI-CWA | **99.60** | **99.90** | **99.80** | **99.50** | **99.60** |
| $R_{ens}$ | Naseer et al. (2021) | 90.43 | 94.39 | 96.67 | 95.48 | 92.63 |
| | MI-CWA | **99.50** | **99.70** | **100.00** | **100.00** | **99.90** |

Table C.2: **Black-box attack success rate (%, ↑) on NIPS2017 dataset.** Surrogates are ResNet18, ResNet32, ResNet50 and ResNet101.

| Method | Backbone | FGSM | BIM | MI | DI | TI | VMI | SVRE | PI | SSA | RAP | MI-SAM | MI-CSE | MI-CWA | SSA-CWA |
|---|---|---|---|---|---|---|---|---|---|---|---|---|---|---|---|
| Normal | AlexNet | 77.6 | 69.2 | 74.9 | 79.2 | 78.2 | 85.3 | 80.9 | 82.6 | 89.7 | 81.9 | 79.5 | 83.1 | 82.9 | **92.7** |
| | VGG-16 | 71.3 | 91.0 | 90.8 | 95.8 | 84.7 | 97.0 | 97.3 | 91.0 | 99.1 | 94.1 | 96.7 | 99.1 | 99.4 | **100.0** |
| | GoogleNet | 58.8 | 88.4 | 87.7 | 95.4 | 79.6 | 96.7 | 96.5 | 88.3 | 98.8 | 90.5 | 94.2 | 98.3 | 98.4 | **99.9** |
| | Inception-V3 | 58.7 | 79.0 | 81.6 | 91.8 | 77.1 | 94.1 | 92.2 | 80.9 | 97.0 | 84.0 | 88.1 | 92.4 | 91.3 | **98.7** |
| | ResNet-152 | 60.2 | 96.6 | 96.1 | 96.8 | 89.9 | 98.5 | 99.5 | 96.4 | 99.1 | 96.2 | 99.1 | **100.0** | **100.0** | 100.0 |
| | DenseNet-121 | 63.1 | 96.2 | 95.2 | 97.2 | 90.3 | 98.2 | 99.3 | 96.0 | 99.4 | 95.0 | 99.0 | 99.8 | 99.8 | **100.0** |
| | SqueezeNet | 86.4 | 89.8 | 90.2 | 95.7 | 86.7 | 96.2 | 96.7 | 93.2 | 98.6 | 93.7 | 94.8 | 98.0 | 98.6 | **99.8** |
| | ShuffleNet-V2 | 83.1 | 78.1 | 83.3 | 86.8 | 78.8 | 91.2 | 89.4 | 85.1 | 95.1 | 89.1 | 88.1 | 91.5 | 91.8 | **97.8** |
| | MobileNet-V3 | 60.6 | 65.6 | 69.5 | 80.6 | 77.3 | 89.7 | 78.2 | 80.4 | 92.2 | 77.1 | 78.2 | 80.2 | 79.9 | **95.7** |
| | EfficientNet-B0 | 55.6 | 89.3 | 87.7 | 94.7 | 77.9 | 96.8 | 96.9 | 86.6 | 98.8 | 92.5 | 95.5 | 98.1 | 97.9 | **99.9** |
| | MNasNet | 66.8 | 87.1 | 84.8 | 93.5 | 75.6 | 96.4 | 95.0 | 84.3 | 98.1 | 92.1 | 94.8 | 97.5 | 97.0 | **99.9** |
| | RegNetX-400MF | 60.3 | 87.9 | 86.8 | 94.9 | 86.0 | 97.2 | 94.9 | 89.3 | 98.7 | 91.3 | 94.4 | 97.6 | 97.5 | **100.0** |
| | ConvNeXt-T | 42.7 | 80.8 | 77.6 | 88.2 | 57.0 | 94.0 | 87.6 | 69.3 | 94.9 | 86.5 | 89.4 | 90.2 | 87.9 | **97.4** |
| | ViT-B/16 | 37.0 | 51.7 | 52.9 | 64.3 | 53.8 | **81.7** | 60.5 | 55.8 | 81.4 | 50.2 | 61.9 | 48.8 | 46.6 | 71.7 |
| | Swin-S | 36.0 | 62.3 | 60.3 | 72.4 | 39.4 | 83.8 | 70.4 | 48.9 | **84.2** | 61.5 | 70.8 | 59.5 | 58.2 | 80.7 |
| | MaxViT-T | 34.1 | 63.0 | 61.6 | 73.1 | 31.9 | 85.1 | 68.6 | 43.9 | **86.7** | 58.9 | 70.5 | 56.7 | 54.5 | 79.0 |
| FGSMAT | Inception-V3 | 55.3 | 51.2 | 55.2 | 59.2 | 65.6 | 73.9 | 61.1 | 66.2 | **84.4** | 59.6 | 58.6 | 60.1 | 60.3 | 78.2 |
| EnsAT | IncRes-V2 | 35.6 | 37.6 | 38.6 | 51.1 | 56.7 | 66.7 | 41.8 | 52.5 | **74.7** | 35.4 | 39.9 | 38.0 | 38.2 | 59.9 |
| FastAT | ResNet-50 | 45.2 | 42.3 | 44.6 | 44.2 | 46.7 | 47.5 | 45.3 | 48.5 | **50.4** | 46.6 | 45.3 | 46.2 | 46.2 | 49.6 |
| PGDAT | ResNet-50 | 35.9 | 31.2 | 34.1 | 35.1 | 38.4 | 41.3 | 35.7 | 42.0 | **43.7** | 36.5 | 36.1 | 35.7 | 35.7 | 40.3 |
| PGDAT | ResNet-18 | 47.1 | 42.0 | 45.4 | 45.3 | 47.6 | 47.2 | 45.6 | 50.4 | 50.6 | 47.8 | 45.8 | 46.8 | 46.9 | **50.7** |
| | WRN-50-2 | 28.1 | 22.7 | 25.6 | 27.4 | 31.2 | 32.6 | 27.3 | 33.9 | **35.3** | 27.7 | 28.0 | 27.6 | 26.7 | 31.7 |
| PGDAT† | XCiT-M12 | 21.7 | 16.8 | 18.5 | 19.9 | 22.9 | 25.0 | 20.7 | 25.3 | **28.1** | 22.0 | 20.9 | 21.3 | 21.2 | 26.5 |
| | XCiT-L12 | 18.7 | 14.9 | 16.3 | 17.3 | 20.9 | 21.6 | 18.6 | 22.3 | **26.0** | 18.4 | 18.4 | 18.5 | 18.2 | 22.6 |
| HGD | IncRes-V2 | 37.8 | 77.4 | 75.4 | 91.1 | 73.0 | 94.2 | 85.5 | 77.9 | 94.7 | 76.9 | 86.7 | 85.8 | 83.6 | **97.8** |
| R&P | ResNet-50 | 69.7 | 97.5 | 96.8 | 97.9 | 94.3 | 98.8 | 99.7 | 97.7 | 99.6 | 96.9 | 99.3 | **100.0** | **100.0** | 100.0 |
| Bit | ResNet-50 | 73.2 | 98.4 | 98.4 | 97.7 | 96.6 | 99.1 | 99.9 | 99.2 | 99.7 | 98.7 | 99.6 | **100.0** | **100.0** | 100.0 |
| JPEG | ResNet-50 | 70.5 | 98.1 | 97.8 | 97.2 | 96.3 | 98.9 | 98.7 | 98.5 | 99.7 | 98.1 | 99.5 | **100.0** | **100.0** | 100.0 |
| RS | ResNet-50 | 66.7 | 97.2 | 96.7 | 97.3 | 93.1 | 98.7 | 99.5 | 97.0 | 99.5 | 96.6 | 99.4 | **100.0** | **100.0** | 100.0 |
| NRP | ResNet-50 | 41.0 | 90.4 | 77.0 | 66.4 | 76.0 | **82.6** | 80.2 | 34.1 | 73.2 | 25.1 | 68.8 | 40.6 | 38.6 | 36.0 |
| DiffPure | ResNet-50 | 57.7 | 68.6 | 75.0 | 84.1 | 88.6 | 95.9 | 85.8 | 89.4 | 96.0 | 76.4 | 84.9 | 81.2 | 79.5 | **95.0** |

Here, $V_{ens}$ represents the ensemble comprising VGG-11, VGG-13, VGG-16, and VGG-19. $R_{ens}$ denotes the ensemble of ResNet-18, ResNet-50, ResNet-101, and ResNet-152. Similarly, $D_{ens}$ corresponds to the ensemble of DenseNet-121, DenseNet-161, DenseNet-169, and DenseNet-201. In this configuration, surrogate models exhibit high similarity. Therefore, it is effective to assess the algorithm's capacity for generalization to unseen target models, particularly those that differ significantly from the surrogate models.

As shown in Tab. C.1, our method surpasses previous techniques by an average of approximately 20%, underscoring the efficacy of our approach in attacking unseen target models.

## C.3   ATTACKING USING LESS DIVERSE SURROGATE MODELS

To further illustrate the efficacy of the CWA algorithm even with a limited diversity of surrogate models, we conducted additional experiments in this section. Specifically, we limit our selection to models of the ResNet family, employing only ResNet-18, ResNet-32, ResNet-50, and ResNet-101 (He et al., 2016) as surrogate models. As shown in Tab. C.2, our SSA-CWA still achieves superior results than other attackers. These results suggest that even when the diversity of surrogate models is limited, adversarial examples generated using the CWA attacker are less prone to overfitting to those surrogate models and continue to generalize effectively to previously unseen target models.

## C.4   ATTACKING USING THE SURROGATE IN DONG ET AL. (2018)

We also evaluate our methods using surrogate models from Dong et al. (2018), which include Inc-v3, Inc-v4, IncRes-v2, and Res-152 from TensorFlow model garden (Yu et al., 2020). Note that these models are different from the corresponding models in TorchVision. This configuration ensures that the majority of the target models are dissimilar to the surrogate models, providing a robust assessment of our algorithm's capability to generate adversarial examples that effectively transfer to diverse and previously unseen models.

As demonstrated in Tab. C.3, SSA-CWA outperforms previous methods significantly when attacking both normally trained models and defended models. Notably, our SSA-CWA achieves a remarkable

Table C.3: **Black-box attack success rate (%,↑) on NIPS2017 dataset.** Surrogate models are Inc-v3, Inc-v4, IncRes-v2 and Res-152 from Dong et al. (2018).

| Method | Backbone | FGSM | BIM | MI | DI | TI | VMI | SVRE | PI | SSA | RAP | MI-SAM | MI-CSE | MI-CWA | SSA-CWA |
|---|---|---|---|---|---|---|---|---|---|---|---|---|---|---|---|
| Normal | AlexNet | 75.8 | 64.2 | 71.3 | 73.3 | 73.5 | 77.6 | 74.8 | 80.2 | 80.1 | 80.4 | 73.7 | 76.8 | 77.8 | **86.0** |
| | VGG-16 | 75.3 | 79.8 | 82.6 | 90.8 | 73.9 | 93.0 | 89.5 | 85.2 | 94.9 | 94.1 | 88.9 | 96.0 | 96.3 | **99.2** |
| | GoogleNet | 61.3 | 78.3 | 79.4 | 91.5 | 74.5 | 94.3 | 87.9 | 84.0 | 96.1 | 93.3 | 88.1 | 96.1 | 95.3 | **99.4** |
| | Inception-V3 | 70.6 | 91.5 | 92.3 | 97.3 | 90.1 | 97.4 | 94.4 | 94.6 | 97.8 | 96.0 | 95.8 | 97.6 | 97.5 | **99.7** |
| | ResNet-152 | 54.9 | 81.4 | 80.3 | 91.9 | 71.1 | 95.6 | 87.0 | 82.0 | 96.3 | 93.1 | 89.8 | 94.7 | 95.4 | **99.3** |
| | DenseNet-121 | 64.1 | 85.7 | 85.4 | 94.7 | 80.6 | 95.8 | 91.3 | 89.2 | 97.0 | 94.9 | 92.4 | 97.4 | 97.1 | **99.7** |
| | SqueezeNet | 85.9 | 80.1 | 84.3 | 89.6 | 78.5 | 90.3 | 89.0 | 87.1 | 92.9 | 92.8 | 87.9 | 91.9 | 93.3 | **97.6** |
| | ShuffleNet-V2 | 81.1 | 71.3 | 76.5 | 81.3 | 69.8 | 82.8 | 81.9 | 81.2 | 84.4 | 85.7 | 80.2 | 84.9 | 84.7 | **91.4** |
| | MobileNet-V3 | 63.6 | 60.1 | 65.0 | 74.0 | 71.1 | 82.4 | 72.2 | 75.5 | 86.5 | 77.9 | 70.1 | 76.3 | 77.2 | **91.1** |
| | EfficientNet-B0 | 60.3 | 77.3 | 77.9 | 90.9 | 68.2 | 94.4 | 86.7 | 80.1 | 95.7 | 93.9 | 88.1 | 93.9 | 93.6 | **99.2** |
| | MNasNet | 67.1 | 70.3 | 74.2 | 87.0 | 63.9 | 89.6 | 84.7 | 72.8 | 92.0 | 91.0 | 81.8 | 91.7 | 90.8 | **98.3** |
| | RegNetX-400MF | 63.6 | 72.1 | 75.8 | 86.3 | 70.3 | 90.6 | 85.1 | 79.3 | 94.6 | 91.2 | 83.9 | 91.1 | 92.1 | **98.7** |
| | ConvNeXt-T | 45.2 | 71.9 | 71.8 | 85.3 | 47.5 | 91.6 | 77.0 | 64.8 | 93.0 | 89.2 | 83.2 | 84.4 | 84.3 | **96.2** |
| | ViT-B/16 | 37.8 | 47.1 | 51.6 | 60.4 | 49.1 | 79.6 | 57.4 | 55.9 | **82.3** | 56.8 | 60.3 | 53.4 | 53.0 | 81.3 |
| | Swin-S | 37.4 | 54.4 | 58.2 | 71.9 | 38.0 | 82.0 | 63.6 | 47.5 | **85.2** | 65.9 | 65.4 | 60.4 | 59.1 | 84.2 |
| | MaxViT-T | 35.9 | 57.6 | 60.0 | 74.0 | 29.9 | 83.6 | 64.4 | 42.0 | **87.1** | 66.0 | 68.9 | 63.9 | 61.7 | 86.9 |
| FGSMAT | Inception-V3 | 61.2 | 54.8 | 57.1 | 62.4 | 68.7 | 73.8 | 61.9 | 68.1 | 79.9 | 63.3 | 60.9 | 63.2 | 63.4 | **80.2** |
| EnsAT | IncRes-V2 | 36.0 | 39.8 | 40.7 | 57.4 | 61.0 | 72.5 | 45.4 | 55.7 | **79.9** | 39.1 | 44.1 | 43.0 | 44.5 | 74.3 |
| FastAT | ResNet-50 | 44.8 | 41.4 | 43.2 | 43.6 | 44.5 | 44.6 | 45.0 | 46.9 | 47.6 | 46.2 | 44.9 | 46.7 | 45.1 | **48.2** |
| PGDAT | ResNet-50 | 35.5 | 30.2 | 32.7 | 33.8 | 36.2 | 37.1 | 34.8 | 39.3 | **41.6** | 36.6 | 34.2 | 32.3 | 35.1 | 39.8 |
| PGDAT | ResNet-50 | 46.5 | 40.8 | 43.6 | 43.9 | 45.1 | 45.6 | 45.1 | 47.3 | 47.6 | 47.3 | 45.2 | 45.9 | 46.0 | **49.1** |
| | WRN-50-2 | 27.5 | 22.3 | 25.2 | 25.9 | 29.2 | 29.7 | 26.8 | 31.4 | **33.1** | 28.8 | 26.8 | 26.7 | 26.8 | 31.5 |
| PGDAT† | XCiT-M12 | 21.1 | 17.1 | 18.8 | 19.7 | 22.1 | 24.5 | 20.6 | 24.5 | **29.0** | 21.7 | 19.8 | 20.4 | 20.1 | 26.9 |
| | XCiT-L12 | 18.9 | 15.0 | 18.0 | 19.5 | 19.3 | 21.8 | 18.5 | 21.1 | **26.3** | 19.3 | 19.1 | 18.2 | 17.0 | 22.9 |
| HGD | IncRes-V2 | 45.7 | 82.3 | 81.9 | 93.2 | 78.4 | 96.1 | 84.3 | 86.4 | 96.4 | 90.7 | 90.3 | 93.1 | 92.4 | **99.5** |
| R&P | ResNet-50 | 65.3 | 79.8 | 80.6 | 93.1 | 74.9 | 93.8 | 86.0 | 84.0 | 95.6 | 92.6 | 87.5 | 93.8 | 94.6 | **99.1** |
| Bit | ResNet-50 | 64.8 | 78.0 | 81.3 | 92.0 | 72.6 | 94.7 | 87.1 | 83.1 | 96.4 | 93.6 | 88.9 | 94.4 | 94.8 | **99.3** |
| JPEG | ResNet-50 | 61.2 | 78.5 | 79.7 | 90.5 | 77.0 | 94.2 | 88.8 | 82.8 | 95.8 | 92.1 | 87.6 | 92.1 | 93.3 | **99.2** |
| RS | ResNet-50 | 61.3 | 81.1 | 80.8 | 91.8 | 74.3 | 94.2 | 88.6 | 83.1 | 95.8 | 93.5 | 89.8 | 94.6 | 94.9 | **99.0** |
| NRP | ResNet-50 | 10.0 | 36.6 | 23.2 | 29.3 | 37.0 | 37.7 | 20.9 | 11.4 | **33.0** | 8.9 | 15.6 | 16.1 | 14.1 | 16.3 |
| DiffPure | ResNet-50 | 52.3 | 51.9 | 62.0 | 72.9 | 73.5 | 86.2 | 67.5 | 76.8 | 89.9 | 70.2 | 68.8 | 67.1 | 67.9 | **94.6** |

94.6% attack success rate against challenging DiffPure defenses (Nie et al., 2022). This underscores the effectiveness of our approach in targeting state-of-the-art defense mechanisms.

## C.5 EXPERIMENTS ON $\epsilon = 4/255$

Many adversarially trained models and defenses are predominantly examined within the context of the $\epsilon = 4/255$ threat model. To remain consistent with these evaluations, we have also carried out an additional experiment under this specific perturbation budget, $\epsilon = 4/255$.

As demonstrated in Tab. C.4, our methods still achieve superior results on most target models. Notably, when attacking the adversarially trained models, our methods outperform previous methods by about 5% on average. This demonstrates the strong efficacy of our methods when attacking with a small perturbation budget, especially adversarially trained models.

## C.6 VISUALIZATION OF ADVERSARIAL PATCHES

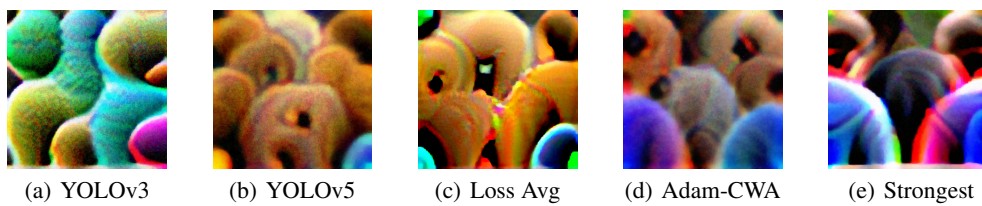

|  (a) YOLOv3 |  (b) YOLOv5 |  (c) Loss Avg |  (d) Adam-CWA |  (e) Strongest |

Figure C.1: **Visualization of adversarial patches from different methods.** The patch simply trained by loss ensemble looks like the fusion of those trained by YOLOv3 and YOLOv5. Adam-CWA captures the common weakness of YOLOv3 and YOLOv5, and therefore generates an completely different patch.

As illustrated in Fig. C.1, we observe that YOLOv5 is more vulnerable to adversarial attacks compared to YOLOv3. Consequently, the patch generated by the Loss Ensemble method resembles the one

Table C.4: **Black-box attack success rate (%, ↑) on NIPS2017 dataset**. Settings are same as Sec. 4.1, except for $\epsilon$, which is set to $4/255$.

| Method | Backbone | FGSM | BIM | MI | DI | TI | VMI | SVRE | PI | SSA | RAP | MI-SAM | MI-CSE | MI-CWA | VMI-CWA | SSA-CWA |
|---|---|---|---|---|---|---|---|---|---|---|---|---|---|---|---|---|
| Normal | AlexNet | 43.4 | 44.2 | 45.8 | 50.3 | 47.6 | 48.5 | 48.9 | 47.8 | 55.3 | 47.7 | 49.5 | 54.9 | 54.8 | 56.6 | **59.8** |
| | VGG-16 | 39.0 | 62.2 | 67.0 | 78.7 | 50.5 | 76.4 | 71.5 | 60.4 | 72.1 | 55.6 | 76.8 | 79.3 | 76.5 | **83.1** | 77.4 |
| | GoogleNet | 35.9 | 49.3 | 54.8 | 72.8 | 41.8 | 65.4 | 58.8 | 50.8 | 70.1 | 45.4 | 63.2 | 68.4 | 64.0 | 73.4 | **76.8** |
| | Inception-V3 | 35.1 | 43.3 | 47.5 | 62.2 | 42.6 | 55.6 | 53.5 | 46.1 | 65.8 | 44.6 | 54.7 | 63.1 | 60.3 | 65.9 | **71.1** |
| | ResNet-152 | 41.7 | 79.7 | 83.7 | 83.7 | 57.6 | 90.0 | 84.5 | 75.3 | 75.8 | 59.1 | 90.4 | 90.3 | 87.3 | **93.7** | 79.8 |
| | DenseNet-121 | 42.2 | 72.1 | 78.7 | 82.4 | 62.1 | 86.5 | 81.2 | 71.6 | 77.2 | 57.4 | 87.7 | 87.5 | 84.5 | **90.8** | 81.7 |
| | SqueezeNet | 49.2 | 63.3 | 67.5 | 76.1 | 56.7 | 74.4 | 73.6 | 63.3 | 76.4 | 60.7 | 73.3 | 80.3 | 79.3 | 83.7 | **84.3** |
| | ShuffleNet-V2 | 44.9 | 49.6 | 53.8 | 59.7 | 47.2 | 57.3 | 57.2 | 52.4 | 63.3 | 50.6 | 56.7 | 65.9 | 64.8 | 67.8 | **70.4** |
| | MobileNet-V3 | 33.3 | 39.9 | 42.7 | 52.2 | 46.4 | 49.5 | 44.5 | 46.8 | 56.6 | 43.3 | 47.9 | 55.2 | 54.9 | 59.7 | **65.5** |
| | EfficientNet-B0 | 29.8 | 49.7 | 57.7 | 71.2 | 37.3 | 67.9 | 55.1 | 48.2 | 62.6 | 44.3 | 66.3 | 69.0 | 64.8 | **76.2** | 69.9 |
| | MNasNet | 37.6 | 58.2 | 62.4 | 75.9 | 41.7 | 73.3 | 63.5 | 53.4 | 65.0 | 51.9 | 71.9 | 73.3 | 70.5 | **77.6** | 71.4 |
| | RegNetX-400MF | 37.0 | 56.6 | 63.4 | 75.3 | 51.7 | 74.3 | 67.1 | 59.0 | 71.7 | 51.7 | 72.5 | 79.9 | 75.9 | **83.7** | 82.4 |
| | ConvNeXt-T | 19.2 | 34.6 | 40.4 | 55.3 | 18.8 | **50.9** | 34.9 | 27.8 | 34.7 | 31.3 | 48.7 | 41.9 | 35.8 | 46.0 | 32.0 |
| | ViT-B/16 | 16.0 | 16.2 | 20.5 | 24.5 | 21.6 | 26.5 | 18.0 | 23.1 | 21.8 | 20.0 | 24.0 | 25.6 | 24.6 | **26.8** | 23.4 |
| | Swin-S | 16.5 | 21.2 | 26.8 | 38.1 | 16.2 | **31.8** | 24.0 | 20.3 | 24.1 | 23.9 | 32.1 | 27.3 | 24.8 | 29.1 | 24.2 |
| | MaxViT-T | 17.1 | 21.7 | 26.3 | 39.3 | 11.2 | **32.8** | 22.5 | 18.4 | 23.4 | 23.1 | 32.2 | 25.5 | 22.4 | 28.6 | 23.7 |
| FGSMAT | Inception-V3 | 35.5 | 35.5 | 38.1 | 40.2 | 40.8 | 39.9 | 39.3 | 38.9 | 50.6 | 40.3 | 40.0 | 51.1 | 50.1 | 50.9 | **56.3** |
| EnsAT | IncRes-V2 | 21.0 | 21.2 | 22.0 | 26.8 | 28.1 | 24.7 | 23.1 | 24.9 | 32.4 | 23.5 | 23.8 | 31.0 | 31.3 | 30.8 | **36.7** |
| FastAT | ResNet-50 | 38.8 | 39.3 | 39.3 | 39.8 | 40.2 | 39.5 | 39.9 | 40.1 | 42.2 | 39.9 | 39.9 | 45.6 | **46.1** | **46.1** | 45.4 |
| PGDAT | ResNet-50 | 25.7 | 26.7 | 27.4 | 28.0 | 28.3 | 27.9 | 27.9 | 27.9 | 30.2 | 27.9 | 27.9 | 36.7 | **36.7** | 36.2 | 33.7 |
| PGDAT | ResNet-18 | 38.3 | 38.6 | 39.0 | 39.5 | 40.8 | 39.1 | 39.6 | 40.0 | 41.7 | 39.6 | 39.5 | 45.9 | **45.8** | 45.6 | 45.7 |
| | WRN-50-2 | 17.9 | 18.4 | 18.7 | 19.7 | 20.0 | 18.9 | 19.2 | 19.4 | 20.9 | 19.9 | 19.4 | 28.1 | **28.1** | 27.4 | 25.5 |
| PGDAT† | XCiT-M12 | 13.3 | 13.5 | 14.0 | 15.0 | 15.5 | 14.9 | 15.1 | 15.2 | 15.4 | 15.2 | 14.8 | 25.5 | **25.6** | 24.9 | 19.9 |
| | XCiT-L12 | 12.5 | 13.4 | 14.0 | 14.0 | 14.7 | 14.4 | 14.8 | 15.0 | 15.2 | 15.2 | 14.6 | 23.3 | **22.9** | 22.3 | 18.8 |
| HGD | IncRes-V2 | 18.7 | 26.8 | 31.9 | **50.8** | 30.7 | 43.4 | 26.6 | 32.3 | 31.8 | 24.5 | 39.0 | 34.0 | 29.8 | 38.7 | 33.5 |
| R&P | ResNet-50 | 51.9 | 86.3 | 89.7 | 91.4 | 76.6 | 93.1 | 93.5 | 85.5 | 85.4 | 67.8 | 93.0 | 94.8 | 93.8 | **95.9** | 89.0 |
| Bit | ResNet-50 | 56.8 | 95.0 | 95.9 | 88.1 | 82.5 | 96.5 | 98.0 | 92.0 | 91.0 | 81.4 | 97.1 | 99.0 | 99.3 | **99.8** | 95.6 |
| JPEG | ResNet-50 | 52.4 | 82.3 | 88.3 | 82.4 | 83.0 | 93.9 | 88.5 | 86.9 | 83.0 | 66.0 | 93.3 | 90.8 | 89.6 | **97.3** | 87.9 |
| RS | ResNet-50 | 47.2 | 86.3 | 89.1 | 86.8 | 67.3 | 92.7 | 91.0 | 82.3 | 83.5 | 68.5 | 93.0 | 94.8 | 93.4 | **96.2** | 88.5 |
| NRP | ResNet-50 | 47.0 | 78.7 | 81.9 | 76.1 | 65.6 | 87.3 | 86.0 | 72.0 | 72.6 | 60.7 | 88.3 | 78.0 | 76.7 | **83.6** | 69.6 |
| DiffPure | ResNet-50 | 23.9 | 22.0 | 25.6 | 33.1 | **40.4** | 28.8 | 23.6 | 35.2 | 31.5 | 26.4 | 28.2 | 28.7 | 29.0 | 30.8 | 33.2 |

obtained by YOLOv5, resulting in similar performance for both patches. We hypothesize that the Loss Ensemble method does not attack the common weakness of YOLOv3 and YOLOv5, and instead nearly solely relies on information from YOLOv5. Contrarily, our proposed method, aims to exploit this common vulnerability and generates a patch that differs significantly from both YOLOv3 and YOLOv5. As a result, our patch are more effective in attacking the object detectors.

In order to craft the strongest universal adversarial patch for object detectors, we ensemble all the models in Huang et al. (2023) and craft an adversarial patch by our CWA algorithm. The patch is visualized in Fig. 1(e). Our patch outperforms previous patches by a large margin. Compared to the previous state-of-the-art methods (*i.e.*, the loss ensemble by enhanced baseline in Huang et al. (2023)), our approach improves by 4.26%, achieving 4.69% mAP on eight testing models in Tab. 2.

## C.7 ABLATION STUDIES

In this section, we investigate the roles of the two additional hyperparameters: the reverse step size $r$ and the inner step size $\beta$. We use the same experimental settings as Sec. 4.1, and average the attack success rates across all target models.

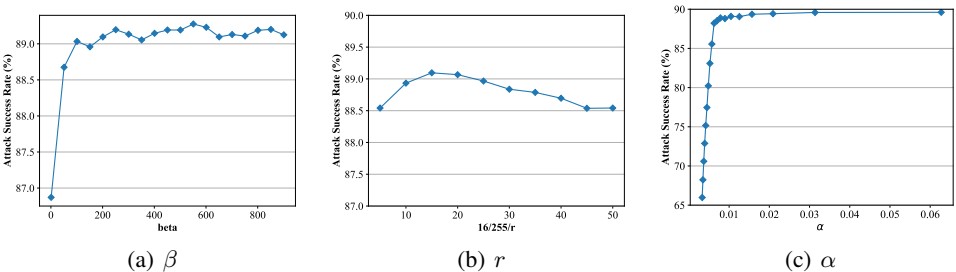

(a) $\beta$          (b) $r$          (c) $\alpha$

**Ablation study on inner step size $\beta$.** Our MI-CSE algorithm could be viewed as optimizing the original loss using the learning rate $\beta$ and the cosine similarity regularization term using the learning rate $\frac{\beta^2}{2}$. As shown in Fig. 2(a), as $\beta$ increases, the proportion of the regularization term gradually

Table C.5: **Black-box attack success rate (%, ↑)** of our methods without incorporating with MI.

| Method | Backbone | FGSM | BIM | MI | SAM | CSE | CWA |
|---|---|---|---|---|---|---|---|
| Normal | AlexNet | 76.4 | 54.9 | 73.2 | 69.5 | 88.1 | 87.3 |
| | VGG-16 | 68.9 | 86.1 | 91.9 | 94.3 | 92.5 | 87.6 |
| | GoogleNet | 54.4 | 76.6 | 89.1 | 92.5 | 92.8 | 88.5 |
| | Inception-V3 | 54.5 | 64.9 | 84.6 | 84.4 | 90.4 | 88.4 |
| | ResNet-152 | 54.5 | 96.0 | 96.6 | 98.0 | 95.3 | 89.1 |
| | DenseNet-121 | 57.4 | 93.0 | 95.8 | 97.7 | 95.5 | 88.1 |
| | SqueezeNet | 85.0 | 80.4 | 89.4 | 91.3 | 94.6 | 91.2 |
| | ShuffleNet-V2 | 81.2 | 65.3 | 79.9 | 82.4 | 91.9 | 90.9 |
| | MobileNet-V3 | 58.9 | 55.6 | 71.8 | 73.1 | 90.6 | 89.7 |
| | EfficientNet-B0 | 50.8 | 80.2 | 90.1 | 93.6 | 93.3 | 87.6 |
| | MNasNet | 64.1 | 80.8 | 88.8 | 93.0 | 90.9 | 87.2 |
| | RegNetX-400MF | 57.1 | 81.1 | 89.3 | 92.9 | 93.6 | 90.1 |
| | ConvNeXt-T | 39.8 | 68.6 | 81.6 | 86.7 | 83.0 | 72.5 |
| | ViT-B/16 | 33.8 | 35.0 | 59.2 | 55.8 | 84.5 | 83.2 |
| | Swin-S | 34.0 | 48.2 | 66.0 | 69.6 | 80.2 | 72.0 |
| | MaxViT-T | 31.3 | 49.7 | 66.1 | 71.1 | 78.5 | 69.0 |
| FGSMAT | Inception-V3 | 53.9 | 43.4 | 55.9 | 55.7 | 88.5 | 89.8 |
| EnsAT | IncRes-V2 | 32.5 | 28.5 | 42.5 | 40.3 | 82.1 | 83.4 |
| FastAT | ResNet-50 | 45.6 | 41.6 | 45.7 | 46.0 | 72.2 | 74.7 |
| PGDAT | ResNet-50 | 36.3 | 30.9 | 37.4 | 36.5 | 70.2 | 72.9 |
| PGDAT | ResNet-18 | 46.8 | 41.0 | 45.7 | 43.6 | 70.8 | 73.6 |
| | WRN-50-2 | 27.7 | 20.9 | 27.8 | 26.4 | 64.7 | 68.1 |
| PGDAT$^\dagger$ | XCiT-M12 | 23.0 | 16.4 | 22.8 | 22.8 | 73.0 | 77.7 |
| | XCiT-L12 | 19.8 | 15.7 | 19.8 | 20.5 | 67.1 | 72.0 |
| HGD | IncRes-V2 | 36.0 | 78.0 | 76.2 | 82.8 | 86.4 | 81.9 |
| R&P | ResNet-50 | 67.9 | 95.8 | 96.3 | 98.3 | 95.0 | 88.7 |
| Bit | ResNet-50 | 69.1 | 97.0 | 97.3 | 98.8 | 97.5 | 89.3 |
| JPEG | ResNet-50 | 68.5 | 96.0 | 96.3 | 98.6 | 96.1 | 89.0 |
| RS | ResNet-50 | 60.9 | 96.1 | 95.6 | 98.2 | 95.8 | 89.9 |
| NRP | ResNet-50 | 36.6 | 88.7 | 72.4 | 92.4 | 82.9 | 62.7 |
| DiffPure | ResNet-50 | 50.9 | 68.5 | 76.0 | 67.9 | 83.3 | 82.8 |

increases, leading to an increase in the attack success rate. However, when $\beta$ becomes too large, the error of our algorithm also increases, causing the attack success rate to plateau or decrease. Therefore, we need to choose an appropriate value of $\beta$ to balance between the effectiveness of the regularization term and the overall performance of the algorithm.

**Ablation study on reverse step size $r$.** Fig. 2(b) shows that the attack success rate initially increases and then decreases as $16/255/r$ increases. This is because when the reverse step size is too large, the optimization direction is opposite to the forward step, leading to a decrease in the attack success rate. Thus, decreasing the reverse step size initially increases the attack success rate. As the reverse step size continues to decrease, MI-CWA gradually degrade to MI-CSE, causing the attack success rate to converge to MI-CSE.

**Ablation study on step size $\alpha$.** We evaluate the average attack success rate for various $\alpha$, ranging from 16/255/1 to 16/255/20. As depicted in Fig. 2(c), When $\alpha < 0.01$, the reverse step size surpass the forward step size. This results in opposing optimization directions and subsequently causes a sharp decline in the efficacy of our method. For $\alpha > 0.01$, the attack success rate remains relatively consistent regardless of specific value of $\alpha$. This showcases our method's resilience to hyper-parameter variations.

**Ablation study on momentum.** We also conduct an experiment where our methods are not combined with MI-FGSM. It is important to note that in other experiments, our methods and the baselines we compare against, except for BIM and FGSM, are combined with MI-FGSM. As shown in Tab. C.5, our methods still achieve superior results than FGSM and BIM. However, it is notable that there is a significant decrease in performance compared with the results that incorporate MI-FGSM. This experiment demonstrates that MI-FGSM has become a popular plug-and-play module capable of efficiently enhancing the performance of various attack algorithms, and it has even become a fundamental and indivisible component in the development of advanced attack algorithms.

# D ANALYSIS AND DISCUSSIONS

In this section, we conduct some additional experiments to prove our claims about the effectiveness of our methods, regarding flattening the loss landscapes and maximizing the cosine similarity of gradients to boost the transferability.

## D.1 VERIFICATION OF THE APPROXIMATION IN CSE

To verify whether $\boldsymbol{I} - \frac{\boldsymbol{g}_i \boldsymbol{g}_i^T}{\|\boldsymbol{g}_i\|_2} \approx \boldsymbol{I}$ holds, We first average the value of $\boldsymbol{I} - \frac{\boldsymbol{g}_i \boldsymbol{g}_i^T}{\|\boldsymbol{g}_i\|_2}$ over NIPS17 dataset, and then downsample this matrix to (256, 256), finally we visualize this matrix in the form of heatmap. The result is shown in Fig. D.1.

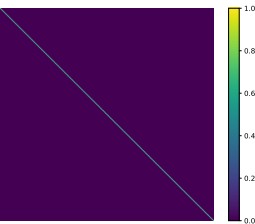

Figure D.1: Heatmap of the matrix.

As shown in the figure, the diagonal elements are close to 1, while the others are close to 0. As a result, the approximation $\boldsymbol{I} - \frac{\boldsymbol{g}_i \boldsymbol{g}_i^T}{\|\boldsymbol{g}_i\|_2} \approx \boldsymbol{I}$ is reasonable. This supports the Assumption A.1.

## D.2 VISUALIZATION OF LANDSCAPE

We illustrate our claim about the relationship between the loss landscape and the transferability of adversarial examples by visualizing the landscapes by different algorithms. In this part, we generate the adversarial example on the ensemble of six surrogate models used in Sec. 4.1 and test on all the defense models in RobustBench (Croce et al., 2021).

For each algorithm, we first craft an adversarial example $\boldsymbol{x}$ via this algorithm, which is expected to be the convergence point during training and near the optimum of each black-box model. Then, we use BIM (Kurakin et al., 2018) to fine-tune the adversarial example on each black-box model to get the position of the optimum of each model $\boldsymbol{p}_i$ (note that this process is white-box). To visualize the loss landscapes of test models around $\boldsymbol{x}$, we perturb $\boldsymbol{x}$ along the unit direction $(\boldsymbol{p}_i - \boldsymbol{x})/\|\boldsymbol{p}_i - \boldsymbol{x}\|_\infty$ for each model and plot the loss curves in one plane. The results of MI and our method MI-CWA are shown Fig. 4(a) and Fig. 4(b) respectively.

As shown in Fig. 4(a) and 4(b), the landscape at the adversarial example crafted by MI-CWA is much flatter than that of MI. It is also noticeable that the optima of the models in MI-CWA are much closer than those in MI. This supports our claim that CWA encourage the flatness of the landscape and closeness between local optima, thus leading to better transferability.

## D.3 COSINE SIMILARITY OF GRADIENTS

We also measure the average cosine similarity of the gradients between training (surrogate) models and testing (black-box) models. The results are shown in Tab. D.1. Our method improves the cosine similarity of gradients both between training models and testing models compared with MI. It shows that our method is more likely to find the common weakness of the training models, and the common weakness of the training models tends to generalize to testing models.

## D.4 TIME COMPLEXITY ANALYSIS

To intuitively illustrate the time complexity of different methods in Sec. 4.1, we list their Number of Function Evaluations (NFEs) in Tab. D.2, where $n$ is the number of surrogate models. As shown, our

Table D.1: **Cosine Similarity of gradients.** MI-CWA universally increases the cosine similarity of gradients among models, regardless of whether they are from $\mathcal{F}_{train}$ or $\mathcal{F}$.

|  | MI | MI-CWA |
|---|---|---|
| $\mathcal{F}_t \times \mathcal{F}_t$ | 0.185 | 0.193 |
| $\mathcal{F}_t \times \mathcal{F}$ | 0.024 | 0.032 |
| $\mathcal{F} \times \mathcal{F}$ | 0.061 | 0.076 |

methods MI-SAM, MI-CSE and MI-CWA are quite efficient. When combined with previous attacks, they maintain their efficiency and enhance their efficacy, showcasing the potential of integrating our approach with leading-edge attacks and optimizers.

We also test the real time cost of generating adversarial examples using different methods, utilizing the surrogate models mentioned in Sec. 4.1. As shown in Tab. D.2, our method incurs only a slight computational overhead compared to the baseline attacks. This demonstrates that our method can serve as a plug-and-play algorithm to effectively and efficiently enhance transfer attack performance.

Table D.2: The number of function evaluations (NFEs) and real time cost of methods that we use in Sec. 4.1.

| Method | FGSM | BIM | MI | DI | TI | VMI | SVRE | PI | SSA | RAP | MI-SAM | MI-CSE | MI-CWA | VMI-CWA | SSA-CWA |
|---|---|---|---|---|---|---|---|---|---|---|---|---|---|---|---|
| NFEs | $n$ | $10n$ | $10n$ | $10n$ | $10n$ | $200n$ | $30n$ | $10n$ | $200n$ | $3400n$ | $20n$ | $10n$ | $20n$ | $210n$ | $210n$ |
| Time (s) | 0.2 | 0.8 | 0.8 | 0.8 | 0.9 | 2.1 | 3.0 | 0.7 | 18.5 | 168.2 | 1.7 | 1.0 | 1.8 | 21.2 | 26.3 |

