# OpenReview forum: "Rethinking Model Ensemble in Transfer-based Adversarial Attacks"
_ICLR.cc/2024/Conference — ICLR 2024 poster_

### Official Review · Reviewer_GTKq · 2023-10-21

**Soundness:** 3 good
**Presentation:** 3 good
**Contribution:** 3 good
**Rating:** 6
**Confidence:** 4

**Summary:**

In this paper, the authors rethink model ensemble in black-box adversarial attacks: exploring the relationship among the transferability of adversarial samples, the Hessian matrix’s F-norm, and the distance between the local optimum of each model and the convergence
point. Based on the above theoretical analysis, the authors define common weaknesses and propose effective algorithms
to find common weaknesses of the model ensemble.

**Strengths:**

- The experiments are solid and compresive: the author consider image classification, object detection and large vision-language models, and the results demonstrate the effectivenss of the proposed method.
- The authors introduce extensive mathmatical proofs, which makes the proposed method more convincing.

**Weaknesses:**

- The description of the experimental part could be improved. Take "Results on normal models" as an example. In the experiment setup, the authors choose MI, VMI and SSA to validate the effectiveness of proposed CWA. However, in "Results on normal models", the authors only discuss MI, and ignore the VMI and SSA. I think the authors should pay more attention on the recent SOTA (e.g., SSA), since MI was proposed in 2018 (5 years ago).

**Questions:**

- How to explain the significant performance degradation of VMI-CWA compared to VMI in Tab 1when the target models are Swin-S and Max ViT-T?

---

> ### Author Response · Authors · 2023-11-15
> **Thank you for the valuable review**
>
> Thank you for appreciating our new contributions as well as providing the valuable feedback. We have uploaded a revision of our paper. Below we address the detailed comments, and hope that you may find our response satisfactory.
>
> ***Question 1: The description of the experimental part could be improved.***
>
> Thank you for the suggestion. We have improved the description of the experimental results and discussed the results of VMI and SSA in the revision. The detailed discussion is as follows.
>
> > By integrating our proposed CWA with recent state-of-the-art attacks VMI and SSA, the resultant attacks VMI-CWA and SSA-CWA achieve a significant level of attacking performance. Typically, SSA-CWA achieves more than 99\% attack success rates for most normal models. VMI-CWA and SSA-CWA can also outperform their vanilla versions VMI and SSA by a large margin. The results not only demonstrate the effectiveness of our proposed method when integrating with other attacks, but also prove the vulnerability of existing image classifiers under strong transfer-based attacks.
>
> ***Question 2: How to explain the significant performance degradation of VMI-CWA compared to VMI in Tab 1 when the target models are Swin-S and Max ViT-T?***
>
> We are sorry that we showed the wrong results of VMI-CWA in Table 1. In our submitted code, we implemented two versions of VMI-CWA. The first one is based on Algorithm 4 in Appendix B.3, in which we adopt the gradients of VMI and perform iterative updates based on our algorithm. The second version is slightly different, which adopts the gradients of our method to perform updates based on VMI. In practice, we observed that the first version performs better, but we showed the results of the second version in Table 1. In the revision, we have corrected the results of VMI-CWA in Table 1. Now the performance of VMI-CWA is better than VMI for the transformers Swin-S and MaxViT-T.

---

> > ### Author Response · Authors · 2023-11-21
> > **Look forward to further feedback**
> >
> > Dear Reviewer GTKq,
> >
> > We thank you again for the valuable comments and appreciation of our contributions. We are looking forward to hearing from you about any further feedback.
> >
> > If you find the response satisfactory, would you like to further increase your rating to highlight this paper at the conference?
> >
> > If you still have questions about our paper, we are willing to discuss them with you and improve our paper.
> >
> > Best, Authors

---

> > ### Comment · Reviewer_GTKq · 2023-11-21
> > **Thanks for your response**
> >
> > Could you present the experimental results of Q1 in a table?

---

> > > ### Author Response · Authors · 2023-11-22
> > > **Thanks for your follow-up comments**
> > >
> > > Thank you for the suggestion. Do you mean to present the results of the two versions of VMI-CWA? If we understand wrongly, please inform us to further address your question.
> > >
> > > For the two versions of VMI-CWA in our initial submission and the revision, the detailed results are:
> > >
> > > | Method | Backbone | VMI-CWA (initial) | VMI-CWA (revision) |
> > > |---------|-------|:--------:| :--------:|
> > > | Normal | AlexNet | 90.8 | 95.9 |
> > > | Normal | VGG-16 | 97.6 | 99.9 |
> > > | Normal | GoogleNet | 96.2 | 99.8 |
> > > | Normal | Inception-V3 | 93.7 | 98.9 |
> > > | Normal | ResNet-152 | 98.1 | 100.0 |
> > > | Normal | DenseNet-121 | 98.8 | 99.9 |
> > > | Normal | SqueezeNet | 97.1 | 99.6 |
> > > | Normal | ShuffleNet-V2 | 94.9 | 98.7 |
> > > | Normal | MobileNet-V3 | 93.2 | 97.8 |
> > > | Normal | EfficientNet-B0 | 96.2 | 99.7 |
> > > | Normal | MNasNet | 95.5 | 99.6 |
> > > | Normal | RegNetX-400MF | 96.5 | 99.8 |
> > > | Normal | ConvNeXt-T | 84.7 | 97.8 |
> > > | Normal | ViT-B/16 | 84.6 | 92.3 |
> > > | Normal | Swin-S | 79.7 | 91.6 |
> > > | Normal | MaxViT-T | 76.0 | 88.1 |
> > > | FGSMAT | Inception-V3 | 88.9 | 91.5 |
> > > | EnsAT | IncRes-V2 | 79.8 | 83.2 |
> > > | FastAT | ResNet-50 | 74.4 | 73.5 |
> > > | PGDAT | ResNet-50 | 71.5 | 72.7 |
> > > | PGDAT | ResNet-18 | 69.4 | 69.2 |
> > > | PGDAT | WRN-50-2 | 63.1 | 63.1 |
> > > | PGDAT$^\dagger$ | XCiT-M12 | 74.3 | 75.1 |
> > > | PGDAT$^\dagger$ | XCiT-L12 | 67.0 | 67.5 |
> > > | HGD | IncRes-V2 | 85.3 | 98.2 |
> > > | R&P | ResNet-50 | 99.8 | 99.8 |
> > > | Bit | ResNet-50 | 99.6 | 100.0 |
> > > | JPEG | ResNet-50 | 99.4 | 100.0 |
> > > | RS | ResNet-50 | 99.1 | 100.0 |
> > > | NRP | ResNet-50 | 95.3 | 33.1 |
> > > | DiffPure | ResNet-50 | 84.3 | 97.3 |
> > >
> > > It can be seen that for most models, the new version of VMI-CWA in the revision performs better than the initial version. The results indicate that using a more appropriate integration (Algorithm 4) of our method with baseline attacks can improve the performance.

---

> > > > ### Comment · Reviewer_GTKq · 2023-11-23
> > > > **Thanks for your reply (2)**
> > > >
> > > > I want to get this results: "Typically, SSA-CWA achieves more than 99% attack success rates for most normal models. VMI-CWA and SSA-CWA can also outperform their vanilla versions VMI and SSA by a large margin"

---

> > > > > ### Author Response · Authors · 2023-11-23
> > > > > **Thanks for pointing out our misunderstanding**
> > > > >
> > > > > Sorry for the misunderstanding of your question. The results are present in Table 1, where we show the attack success rates of various methods including VMI (8th column), SSA (11th column), VMI-CWA (16th column), and SSA-CWA (17th column). From the table, we can observe that VMI-CWA and SSA-CMA achieve better performance than VMI and SSA, and the strongest method SSA-CWA achieves more than 99% attack success rates for most normal models. We would also show the results below to better address your question.
> > > > >
> > > > > | Method | Backbone | VMI | SSA | VMI-CWA | SSA-CWA |
> > > > > |---------|-------|:--------:| :--------:| :--------:| :--------:|
> > > > > | Normal | AlexNet | 83.3 | 89.0 | 95.9 | 96.9 |
> > > > > | Normal | VGG-16 | 94.8 | 97.7 | 99.9 | 99.9 |
> > > > > | Normal | GoogleNet | 94.2 | 97.2 | 99.8 | 99.8 |
> > > > > | Normal | Inception-V3 | 91.1 | 95.6 | 98.9 | 99.6 |
> > > > > | Normal | ResNet-152 | 97.1 | 97.6 | 100.0 | 100.0 |
> > > > > | Normal | DenseNet-121 | 96.6 | 98.2 | 99.9 | 100.0 |
> > > > > | Normal | SqueezeNet | 94.2 | 97.2 | 99.6 | 99.8 |
> > > > > | Normal | ShuffleNet-V2 | 89.9 | 93.9 | 98.7 | 98.8 |
> > > > > | Normal | MobileNet-V3 | 87.3 | 91.4 | 97.8 | 98.1 |
> > > > > | Normal | EfficientNet-B0 | 94.6 | 96.9 | 99.7 | 99.9 |
> > > > > | Normal | MNasNet | 94.1 | 97.2 | 99.6 | 99.9 |
> > > > > | Normal | RegNetX-400MF | 95.3 | 97.4 | 99.8 | 99.9 |
> > > > > | Normal | ConvNeXt-T | 92.4 | 93.1 | 97.8 | 98.1 |
> > > > > | Normal | ViT-B/16 | 81.8 | 83.0 | 92.3 | 90.0 |
> > > > > | Normal | Swin-S | 84.2 | 85.2 | 91.6 | 88.4 |
> > > > > | Normal | MaxViT-T | 83.5 | 85.2 | 88.1 | 86.1 |
> > > > > | FGSMAT | Inception-V3 | 72.3 | 84.3 | 91.5 | 92.7 |
> > > > > | EnsAT | IncRes-V2 | 66.4 | 76.1 | 83.2 | 84.1 |
> > > > > | FastAT | ResNet-50 | 51.4 | 34.7 | 73.5 | 70.4 |
> > > > > | PGDAT | ResNet-50 | 47.1 | 25.3| 72.7 | 66.8 |
> > > > > | PGDAT | ResNet-18 | 48.9 | 41.1 | 69.2 | 65.9 |
> > > > > | PGDAT | WRN-50-2 | 36.2 | 18.7 | 63.1 | 55.6 |
> > > > > | PGDAT$^\dagger$ | XCiT-M12 | 33.4 | 13.1 | 75.1 | 66.3 |
> > > > > | PGDAT$^\dagger$ | XCiT-L12 | 30.8 | 11.5 | 67.5 | 59.4 |
> > > > > | HGD | IncRes-V2 | 92.0 | 93.9 | 98.2 | 98.7 |
> > > > > | R&P | ResNet-50 | 98.7 | 98.9 | 99.8 | 100.0 |
> > > > > | Bit | ResNet-50 | 99.0 | 99.5 | 100.0 | 100.0 |
> > > > > | JPEG | ResNet-50 | 98.6 | 99.2 | 100.0 | 100.0 |
> > > > > | RS | ResNet-50 | 96.9 | 98.1 | 100.0 | 100.0 |
> > > > > | NRP | ResNet-50 | 89.0 | 92.8 | 33.1 | 85.4 |
> > > > > | DiffPure | ResNet-50 | 92.6 | 93.4 | 97.3 | 97.5|

---

> ### Comment · Reviewer_GTKq · 2023-11-23
> **Thanks for your reply (3)**
>
> Thanks for your reply. My concern has been addressed and I will keep the rating, i.e., tend to accept this paper.

---

> > ### Author Response · Authors · 2023-11-23
> > **Thank you for your recognition.**
> >
> > Thank you for your recognition and valuable suggestions. We look forward to receiving more of your insightful feedback to further improve our paper.

---

### Official Review · Reviewer_9Y8Q · 2023-10-31

**Soundness:** 4 excellent
**Presentation:** 3 good
**Contribution:** 3 good
**Rating:** 8
**Confidence:** 3

**Summary:**

This paper studies the transfer-based attack towards the ensemble of models. The authors suggest that the ensemble model would have "common weaknesses" that are strongly correlated with adversarial examples' transferability and propose the common weaknesses attack (CWA). Theoretically, this paper provides clear intuition on why the models have common weaknesses. This paper also conducts extensive experiments to validate their theoretical findings.

**Strengths:**

1. This paper conducts extensive experiments on several datasets, demonstrating that CWA receives superior results than previous methods.
2. The intuition of common weakness is strong and clear. This paper converts the task "crafting adversarial examples on ensemble models" to "optimizing the second term in Equation (2)". By Theorem 3.1, this term is further decomposed into a "flatness term" and a "closeness term". These two terms can be efficiently optimized by SAM and CSE.

**Weaknesses:**

1. The intuition behind SAM and CSE is not clear enough. Equation (4) and (5) is confusing for those readers not familiar with this area.
2. In Section 3.1, the authors mentioned that "the goal of transfer-based attacks is to craft an adversarial example $x$ that is misclassified by all models in $\mathcal{F}$". However, fooling all target models seems to be an impossible job in practice. Besides, the experiments in this paper cannot support this claim.
3. The perturbation in Figure 3 is not imperceptible to humans.

**Questions:**

See the Weakness part.

**Details Of Ethics Concerns:**

I have no ethical concerns.

---

> ### Author Response · Authors · 2023-11-15
> **Thank you for the valuable review**
>
> Thank you for appreciating our new contributions as well as providing the valuable feedback. We have uploaded a revision of our paper. Below we address the detailed comments, and hope that you may find our response satisfactory.
>
> ***Question 1: The intuition behind SAM and CSE is not clear enough.***
>
> Thank you for pointing this out. SAM is an effective method to acquire a flatter landscape, which is formulated as a min-max optimization problem, in a similar way to adversarial training. The inner maximization aims to find a direction along which the loss changes more rapidly;
> while the outer problem minimizes the loss at this direction to improve the flatness of loss landscape. We have added the introduction of SAM in the beginning of Section 3.3.
>
> For CSE, to maximize the cosine similarity between the gradients of different models, we are inspired by Nichol et al. [1] to develop a first-order algorithm. Nichol et al. propose a meta-learning algorithm that iteratively performs gradient updates for each task, and they theoretically show that this procedure can increase the inner product between gradients of different tasks. We extend this method to also increase the cosine similarity between the gradients of different models, with theoretical analyses in Appendix B.2.
>
>
> ***Question 2: Overclaim on fooling all target models.***
>
> Thank you for pointing this out. We have revised our paper to avoid this overclaim. Now the text becomes
>
> > Given a natural image $\boldsymbol{x}_{nat}$ and the corresponding label $y$, transfer-based attacks aim to craft an adversarial example $\boldsymbol{x}$ that could be misclassified by the models in $\mathcal{F}$.
>
> ***Question 3: The perturbation is Figure 3 is not imperceptible to humans.***
>
> We are sorry that the first image in Figure 3 was placed wrongly -- we used the adversarial image with $\epsilon=32/255$, such that this image is more perceptible. In the revision, it is changed to the correct image with $\epsilon=16/255$. The perturbations are visually similar to those generated by other attack methods with the same perturbation budget.
>
> **Reference:**
>
> [1] Nichol et al., On first-order meta-learning algorithms, 2018.

---

> > ### Comment · Reviewer_9Y8Q · 2023-11-15
> > **Thanks for your reply**
> >
> > The authors' reply addressed most of my concerns. The intuition behind SAM and CSE seems adequate to me. I hope that the authors can include them in future revisions so as to improve the overall interpretability of your method.

---

> > > ### Author Response · Authors · 2023-11-21
> > > **Thanks for the update**
> > >
> > > Thank you again for the valuable review. We will try our best to further improve the paper in the final version.

---

### Official Review · Reviewer_JLTP · 2023-11-01

**Soundness:** 3 good
**Presentation:** 2 fair
**Contribution:** 3 good
**Rating:** 6
**Confidence:** 3

**Summary:**

This paper proposes a new method for constructing adversarial examples for black-box models, using an ensemble of surrogate models. Using a second order approximation of the adversarial loss function, the authors construct an upper bound using the Hessian on the adversarial loss, and establish approximate methods for minimizing the upper bound, yielding their proposed attack: Common Weakness Attack (CWA). The CWA algorithm consists of two steps: first a gradient ascent step on the averaged logits, inspired by sharpness-aware minimization, and a second gradient descent step using their proposed cosine similarity encourager (CSE). The authors evaluate their attack method on a variety of model architectures using the NIPS2017 dataset, and compare with a variety of different attack algorithms from the literature.

**Strengths:**

- The paper tackles an important subject, understanding how vulnerable can ML models be is crucial for safe-guarding them against possible adversaries

- The proposed method, as far as I can tell, seems novel to me.

- The strengths of this paper are primarily in the effectiveness of their proposed method, as it seems to combine very well with prior attacks (e.g, MI-CWA, SSA-CWA) and achieves superior results. The transfer-based black-box attacks against Bard is interesting.

- The proposed algorithm seems reasonable to implement

**Weaknesses:**

- I felt that the paper was bit hard to follow. Section 3 mixes a lot of prior results with proposed ones. Thus it makes it a bit hard to distinguish original contributions vs reusing prior results. For instance, it would greatly improve the readability if SAM was properly explained prior to this section (or as a subsection). Figures 1 and 2 are not very informative (see Questions), they lack axis and/or labels. It would also improve the quality of the paper if the authors include a mathematical description of some of the prior ensemble attack methods (e.g. is it simply PGD on the averaged logits?), to better understand the contrast between them and the proposed method.

- Despite the compactness of Algorithm 1, there is a lack of analysis on its complexity. How does it compare (in terms of wall-clock time) to prior attacks or vanilla PGD-style attacks?

**Questions:**

- In Fig 1, it is not clear what is being plotted, and the different colors are not well contrasted.
- In Sec 3.1, below eq (2), the authors write " ... we can see that a smaller value of ... means a smaller test error ...". What is the test error in this case? is it the error on the black-box model? the term test error is rather confusing.
- In Theorem 3.1, there is a typo, should be $||H_i||_F$?
- In Sec 3.2: " There are some researches ..." typo
- In Fig 2, what are $x_t$, $x_t^f$, .... The symbols are not defined.
- In Alg1, does the order of the classifiers in the for loop matter? if so how is it currently chosen?
- In Table 1, some of the highlighted numbers are not the best performing numbers, for instance the FastAT RN50 row.
- Can the proposed method be also applied in the white-box setting? i.e. assume $\mathcal{F} = [f]$ and simply attack the classifier.  Have the authors experimented with something like this?

---

> ### Author Response · Authors · 2023-11-15
> **Thank you for the valuable review**
>
> Thank you for the supportive review. We are encouraged by the appreciation of the significance, novelty, and effectiveness of our method. We have uploaded a revision of our paper. Below we address the detailed comments, and hope that you may find our response satisfactory.
>
>
> ***Question 1: The paper was bit hard to follow.***
>
> Thank you for pointing out the issues. We further improve the writing to address them in the revision. Below are the details.
>
> - **Improve the readability of SAM.** We have provided a brief introduction of SAM in the beginning of Section 3.3, to provide the background knowledge of SAM and explain why it can be used for our problem.
>
> - **Figures 1 and 2 are not very informative.** We have updated Figures 1 and 2 to be more informative, including adding the axes and labels, changing the colors, and specifying the symbols.
>
> - **Include a mathematical description of the prior ensemble attacks.** We have added a new subsection 3.1 in the revision to introduce the backgrounds of adversarial attacks.
>
> We hope that the new revision can be easier to follow. We will further improve the paper in the final.
>
> ***Question 2: Lack of analysis on complexity.***
>
> Thank you for the suggestion. We discussed the computational efficiency in Section 4.4 and Appendix D.4. We further report the actual runtime of different attack methods in Table D.2.  Our method incurs only a slight
> computational overhead compared to the baseline attacks. This demonstrates that our method can
> serve as a plug-and-play algorithm to effectively and efficiently enhance transfer attack performance. As also discussed in Section 4.4, given a fixed computation overhead, we can use a smaller number of attack iterations in our method, which also leads to better results, as shown in Figure 4(c).
>
> ***Question 3: Various typos and minor problems.***
>
> Thank you for pointing them out. We address these issues and carefully proofread the paper. We have uploaded a new revision. Below are the details.
>
> - **What is being plotted in Figure 1.** Figure 1 shows an illustration of common weakness, where we plot the synthetic loss curves of different models w.r.t. the input $\boldsymbol{x}$. We show that the generalization performance is correlated with the flatness of loss landscape and the distance to the local optimum of each model.
>
> - **Typos in Sec. 3.1, Theorem 3.1, Sec. 3.2.** Thank you for pointing them out. We have corrected them in the revision.
>
> - **The symbols in Figure 2 are not defined.** We have revised the caption to link the symbols to the equations in the main text.
>
> - **The order of classifiers in Alg. 1.** The order of classifiers is randomly set and fixed during the optimization process.
>
> - **Some highlighting numbers are not the best in Table 1.** Thank you for pointing this out. We carefully checked the results and marked the best results in bold in the revision.
>
> - **Can the method be applied in the white-box setting.** Yes, our method can be applied in the white-box setting. But when there is only one white-box model, our method would degenerate to PGD or MI since we focus on ensemble-based attacks. Therefore, we further show the attack performance of our method against the six white-box models in the ensemble, as detailed in Section 4.1, and compare with PGD-10 and MI. The results are shown below.
>
> |         | Epsilon | ResNet18 | ResNet34 | ResNet50 | ResNet101 | ResNet50 (Salman et al. (2020)) | XCiT-S12 (Debenedetti et al. (2023)) |
> |---------|:-------:|:--------:|:--------:|:--------:|:---------:|:-------------------------------:|:-----------------------------------:|
> | PGD     | 16/255  |   98.1   |   98.3   |   98.2   |   98.4    |              34.6               |                19.8                 |
> | MI | 16/255  |   98.5   |   98.6   |   98.4   |   98.5    |              41.9               |                27.0                 |
> | MI-CWA  | 16/255  |  100.0   |  100.0   |  100.0   |   100.0   |              91.1               |                92.2                 |
> | PGD     |  4/255  |   96.2   |   95.9   |   96.1   |   96.2    |              27.2               |                15.5                 |
> | MI |  4/255  |   97.0   |   96.8   |   97.1   |   97.4    |              28.0               |                16.2                 |
> | MI-CWA  |  4/255  |  100.0   |  100.0   |  100.0   |   100.0   |              47.0               |                39.1                 |
>
> It can be seen that our method also leads to higher attack success rates over PGD and MI with a small number of attack iterations. It indicates that our method converges much faster than the baselines since it can align the gradients of different models.

---

> > ### Author Response · Authors · 2023-11-21
> > **Look forward to further feedback**
> >
> > Dear Reviewer JLTP,
> >
> > We thank you again for the valuable comments and appreciation of our contributions. We are looking forward to hearing from you about any further feedback.
> >
> > If you find the response satisfactory, would you like to further increase your rating to highlight this paper at the conference?
> >
> > If you still have questions about our paper, we are willing to discuss them with you and improve our paper.
> >
> > Best, Authors

---

### Official Review · Reviewer_s2Ct · 2023-11-01

**Soundness:** 3 good
**Presentation:** 4 excellent
**Contribution:** 3 good
**Rating:** 8
**Confidence:** 2

**Summary:**

The paper proposes a novel adversarial attacks (Common Weakness Attack) against ensembles of model. It composes of two components, sharpness aware minimization (SAM) and cosine similarity encourager (CSE) that regularize the flatness of the loss landscape and the closeness to the optimum respectively. The author performs extensive experiments over extensive datasets and models.

**Strengths:**

1. The paper is well-written in general and has very clear motivation and mathematical formulations. The derivation of attack based on the second order appropriation is intuitive.
2. The author conducts very extensive empirical studies with many choices of architectures and datasets.

**Weaknesses:**

1. In table 1, the author only provides the results of CWA in combintation with other methods. Is there results for CWA only, and how well it performs.
2. As an ablation study, the author might want to try different norm decomposition, e.g. the operator norm of H.
3. The author should clarify the novelty compared to the previous methods. Specially, MI, VMI, SSA are all existing proposed attacks. The sharpness aware minimization techniques have been previously used.

**Questions:**

1. Can the author explain why the model is able to achieve significantly more effective attacks compared the existing methods for the adversarial trained model? I am mainly concerned about the fairness of the evaluation. Also, can the author explain how the AT model trained for ensembles in their experiments?

---

> ### Author Response · Authors · 2023-11-15
> **Thank you for the valuable review**
>
> Thank you for the supportive review. We are encouraged by the appreciation of the novelty, clarity, and thorough experiments of this paper. We have uploaded a revision of our paper. Below we address the detailed comments, and hope that you may find our response satisfactory.
>
> ***Question 1: The performance of CWA only.***
>
> Thank you for the suggestion. We further conduct experiments of SAM, CSE, and CWA without integrating with other methods. The detailed results are shown in Table C.5 in Appendix C.7. The average performance of different methods over all black-box models are shown below.
>
> | Method | FGSM | BIM | MI | SAM | CSE | CWA | MI-SAM | MI-SCE | MI-CWA |
> | :-----: | :-----: | :-----: | :-----: | :-----: | :-----: | :-----: | :-----: | :-----: | :-----: |
> | Attack success rate (\%) | 50.89 | 63.71 | 71.65 | 73.38 | 85.85 | 82.55 | 77.92 | 90.68 | 90.76 |
>
> It can be seen that SAM, CSE, and CWA outperform the baselines FGSM, BIM, and MI by a large margin. However, their performance is inferior to their counterparts with MI, indicating that our methods and MI are orthogonal and can be integrated together to further improve the performance.
>
> ***Question 2: Different norm decomposition.***
>
> Thank you for the suggestion. It is true that we can decompose the second-order term $(\boldsymbol{x}-\boldsymbol{p}_i)^\top \boldsymbol{H}_i (\boldsymbol{x}-\boldsymbol{p}_i)$ with different norms (see Appendix A.1 for a more general result). However, with a different norm of $\boldsymbol{H}_i$, it is hard to interpret the meaning of minimizing this norm, and we can hardly associate it with generalization/transferability of adversarial examples. Besides, we need to calculate the third-order derivative of the loss function w.r.t. input $\boldsymbol{x}$, which would be intractable for deep neural networks. Therefore, it is hard to deal with other norms of $\boldsymbol{H}_i$, and we only focus on the Frobenius norm of  $\boldsymbol{H}_i$ with a clear connection to loss flatness and an alternative method based on sharpness aware minimization for optimization.
>
> ***Question 3: Clarify the novelty compared to the previous methods.***
>
> The main novelty of this paper lies in a new adversarial attack method from the perspective of model ensembling. Specifically, we use a quadratic approximation of the loss function and discover that the second-order term can be upper bounded by the product of the Frobenius norm of the Hessian matrix and the distance to the local optimum of each model. Therefore, we propose a common weakness attack to minimize the upper bound with great effectiveness. The existing attacks (e.g., MI, VMI, SSA) mainly focus on developing optimization algorithms or data augmentation techniques to improve transferability, while do not consider model ensembling in adversarial attacks. The sharpness aware minimization (SAM) is commonly used to train deep neural networks, but we extend it to adversarial attacks to improve the transferability based on our analysis.
>
> ***Question 4: Why is the method more effective for adversarial trained model?***
>
> The main reason is that our method can better exploit the vulnerabilities of different surrogate models in the ensemble by finding their common weakness. As we included two adversarially trained models (which are publicly available by [1,2]) in the ensemble, the adversarial examples generated by our method can better transfer to attack other black-box adversarially trained models. But for baseline attacks that directly average different surrogate models, they tend to attack easier models (i.e., normally trained models) [3], such that the useful gradient information would be overwhelmed by these normal models, leading to inferior results for adversarially trained models.
>
> **Reference:**
>
> [1] Salman et al., Do adversarially robust imagenet models transfer better? NeurIPS 2020.
>
> [2] Debenedett et al., A light recipe to train robust vision
> transformers. IEEE Conference on Secure and Trustworthy Machine Learning, 2023.
>
> [3] Dong et al., Discovering adversarial examples with momentum, https://arxiv.org/abs/1710.06081v1.

---

> > ### Comment · Reviewer_s2Ct · 2023-11-17
> >
> > Thank you for the response. I think it addresses all my concerns.

---

> > > ### Author Response · Authors · 2023-11-21
> > > **Thanks for the update**
> > >
> > > Thank you again for the valuable review. We will try our best to further improve the paper in the final version.

---

### Meta-Review · Area_Chair_SkaB · 2023-12-08

**Metareview:**

This paper studies the transfer-based attack towards the ensemble of models. The authors propose the common weaknesses attack (CWA) to address the common weaknesses that are strongly correlated with adversarial examples transferability. The theretical analysis also support their claims. The experiments are solid. All the reviewers agree the novelty and contribution of this paper and thus recommend the acceptance.

**Justification For Why Not Higher Score:**

the theoretical contribution of this paper is limited.

**Justification For Why Not Lower Score:**

All the reviewers agree to accept this paper.

---

### Decision · Program_Chairs · 2024-01-16

Accept (poster)